# BEP: A Binary Error Propagation Algorithm for Binary Neural Networks Training

**Luca Colombo**[1][*]  **Fabrizio Pittorino**[1]  **Daniele Zambon**[2]

**Carlo Baldassi**[3]  **Manuel Roveri**[1]  **Cesare Alippi**[1,2]

[1]Department of Electronics, Information and Bioengineering, Politecnico di Milano, Italy
[2]Faculty of Informatics and IDSIA, Università della Svizzera italiana, Switzerland
[3]Department of Computing Sciences and BIDSA, Bocconi University, Italy

## Abstract

Binary Neural Networks (BNNs), which constrain both weights and activations to binary values, offer substantial reductions in computational complexity, memory footprint, and energy consumption. These advantages make them particularly well suited for deployment on resource-constrained devices. However, training BNNs via gradient-based optimization remains challenging due to the discrete nature of their variables. The dominant approach, quantization-aware training, circumvents this issue by employing surrogate gradients. Yet, this method requires maintaining latent full-precision parameters and performing the backward pass with floating-point arithmetic, thereby forfeiting the efficiency of binary operations during training. While alternative approaches based on local learning rules exist, they are unsuitable for global credit assignment and for back-propagating errors in multi-layer architectures. This paper introduces Binary Error Propagation (BEP), the first learning algorithm to establish a principled, discrete analog of the backpropagation chain rule. This mechanism enables error signals, represented as binary vectors, to be propagated backward through multiple layers of a neural network. BEP operates entirely on binary variables, with all forward and backward computations performed using only bitwise operations. Crucially, this makes BEP the first solution to enable end-to-end binary training for recurrent neural network architectures. We validate the effectiveness of BEP on both multi-layer perceptrons and recurrent neural networks, demonstrating gains of up to $+6.89\%$ and $+10.57\%$ in test accuracy, respectively. The proposed algorithm is released as an open-source repository.

## 1 Introduction

The design of Neural Networks (NNs) with weights and activations constrained to binary values, typically $\pm 1$, is a promising direction for building models suited to resource-constrained environments. In particular, Binary Neural Networks (BNNs) offer a compelling solution for deploying Deep Learning (DL) models on edge devices and specialized hardware, where computational efficiency, power consumption, and memory footprint are critical design constraints (Courbariaux et al., 2015; Rastegari et al., 2016; Hubara et al., 2016). Their primary advantage lies in replacing costly Floating-Point (FP) arithmetic with lightweight bitwise operations such as XNOR and Popcount, resulting in substantial reductions in computational complexity (Qin et al., 2020; Lucibello et al., 2022).

Despite these advantages, effectively training BNNs remains challenging due to the non-differentiable nature of their activation functions. Consequently, conventional gradient-based learning algorithms cannot be directly applied, leading to two main classes of solutions. The most prevalent is Quantization Aware Training (QAT), which formulates the problem within a continuous optimization framework. In this paradigm, full-precision latent weights are maintained, and non-differentiable activations are bypassed during the backward pass using the Straight-Through Estimator (STE) (Bengio et al., 2013). However, reliance on real-valued computations for gradient calculation and weight updates confines the efficiency of binary arithmetic to the forward pass only (Sayed et al., 2023).

---

[*]Corresponding author. Email: luca2.colombo@polimi.it

An alternative line of research has explored purely binary, gradient-free learning rules, often inspired by principles from statistical physics (Baldassi et al., 2015; Baldassi, 2009). These methods operate directly on binary weights and avoid continuous surrogates. A recent extension applied this approach to binary Multi-Layer Perceptrons (MLPs) by generating local error signals at each layer using fixed random classifiers (Colombo et al., 2025). However, a fundamental limitation of this approach is that credit assignment remains local and error information does not propagate from the final output layer through the NN. This constraint makes such rules inapplicable to architectures where learning depends on end-to-end error propagation across layers, such as Recurrent Neural Networks (RNNs).

From this perspective, this paper addresses the following research question: *Is it possible to formulate a multi-layer, global credit assignment mechanism that back-propagates errors through the NN while operating exclusively within the binary domain?* To the best of our knowledge, we introduce the first *fully binary* error Backpropagation (BP) algorithm capable of effectively training BNNs without relying on FP gradients. The algorithm, called Binary Error Propagation (BEP) hereafter, establishes a binary analog of the standard BP chain rule, where error signals – represented as binary vectors – are computed at the output and propagated backward through each layer of the NN. To ensure learning stability, BEP employs integer-valued hidden weights that provide synaptic inertia and mitigate catastrophic forgetting (Kirkpatrick et al., 2017). Crucially, the *entire forward and backward passes* rely solely on efficient XNOR, Popcount, and increment/decrement operations. In summary, this work makes the following contributions:

- We formalize a fully binary BP algorithm for BNNs that propagates binary-valued error signals end-to-end, establishing a discrete analog of the gradient-based BP chain rule.
- We demonstrate that BEP successfully train both MLP and RNN architectures, overcoming the limitations of prior local and gradient-based learning methods.

The remainder of this paper is organized as follows. Section 2 reviews related work. Section 3 formalizes the proposed BEP algorithm. Section 4 presents experimental results on binary MLP and RNN architectures. Finally, Section 5 draws conclusions and outlines future research directions.

## 2 RELATED LITERATURE

The predominant paradigm for training BNNs is QAT (Courbariaux et al., 2015; Hubara et al., 2016; Rastegari et al., 2016). In this approach, models maintain latent full-precision parameters that are binarized during the forward pass, while gradients are computed with respect to the latent parameters using a surrogate gradient, typically a STE (Bengio et al., 2013). The STE approximates the derivative of the non-differentiable *sign* function as an identity within a bounded region, enabling the use of standard BP. Numerous subsequent works have built upon this foundation, introducing improvements such as learnable representations, enhanced architectures, and strategies to narrow the accuracy gap with full-precision models (Lin et al., 2017; Liu et al., 2020; Tu et al., 2022; Schiavone et al., 2023). While QAT has achieved strong empirical results, it remains fundamentally a continuous optimization method applied to a discrete problem. Training relies on FP arithmetic, which prevents the full realization of BNN efficiency during learning and introduces a discrepancy between training and inference dynamics (Yin et al., 2019). Recent work (Liu et al., 2018; Bulat & Tzimiropoulos, 2019; Vargas et al., 2024) has further refined QAT by incorporating residual connections and improved surrogate-gradient mechanisms to enhance gradient flow. In contrast, our approach diverges from this paradigm by eliminating the need for any real-valued parameters or surrogate gradients.

A distinct line of research frames BNN training as a purely binary optimization problem. Early work in statistical physics explored combinatorial optimization techniques for training single-layer perceptrons (Gardner, 1988; Engel, 2001), while later studies investigated fully binary training via global heuristics such as simulated annealing (Kirkpatrick et al., 1983; Hinton, 1990) and evolutionary strategies (Salimans et al., 2017; Such et al., 2017; Loshchilov & Hutter, 2016). Although these methods avoid continuous relaxations, they explore the weight space via stochastic perturbations (e.g., random weight flips) and lack a structured, layer-wise credit assignment comparable to BP, which limits their scalability and efficiency in deep settings. Our work addresses this limitation by developing a multi-layer, gradient-free training method with a deterministic error-propagation rule.

In parallel, research at the intersection of statistical physics and computational neuroscience has developed efficient *local* learning rules for binary neurons. Algorithms such as the Clipped Perceptron

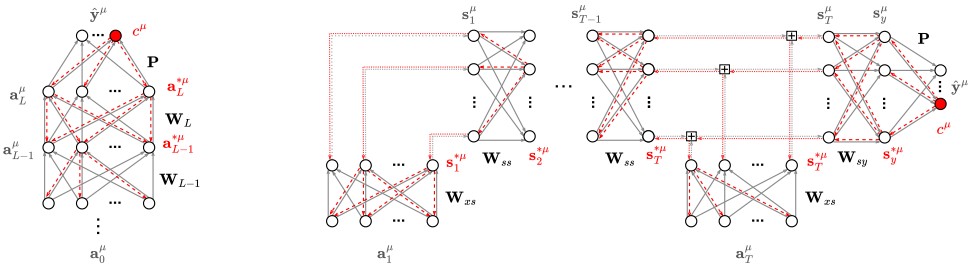

(a) BEP applied to a binary MLP          (b) BEP applied to a binary RNN

Figure 1: Information flow for a sample $\mu$ in an MLP and an RNN trained with BEP. Each model uses a binary core and a fixed classifier. The forward and backward passes are shown in gray and red.

with Reinforcement (CP+R) (Baldassi, 2009) and related message-passing approaches (Baldassi et al., 2007; 2015) introduce integer-valued hidden variables to represent synaptic confidence, demonstrating that single binary units can learn effectively. Multi-layer extensions of these rules generate layer-wise local error signals using fixed random classifiers, enabling training of several binary layers but still lacking end-to-end propagation of task loss (Colombo et al., 2025). This structural constraint notably precludes their application to a relevant class of recurrent sequential architectures, such as RNNs. BEP overcomes these shortcomings by introducing a *global* credit assignment mechanism that propagates binary error signals end-to-end, bridging the gap between fully binary optimization and multi-layer training of deep architectures.

## 3    THE BINARY ERROR PROPAGATION ALGORITHM

In this section, we formalize the BEP algorithm for training multi-layer BNNs using exclusively binary operations. We consider a supervised learning setting defined by a classification task and a corresponding dataset $\mathbf{X} = \{\mathbf{x}^\mu, c^\mu\}_{\mu=1}^N$ of size $N$, where $\mathbf{x}^\mu \in \mathbb{R}^{K_0}$ are input patterns of dimension $K_0$, and $c^\mu \in \{1, \dots, C\}$ are their corresponding labels, with $C$ denoting the number of classes. Specifically, Section 3.1 introduces the BNN architecture and forward-pass dynamics. The binary backward pass for error propagation is described in Section 3.2. The weight-update rule is defined in Section 3.3, while the analogy with standard full-precision BP is presented in Section 3.4. Finally, Section 3.5 demonstrates the application of the proposed BEP algorithm to RNN architectures.

### 3.1   BNN ARCHITECTURE AND FORWARD PROPAGATION

The NN under consideration consists of two main components: a trainable binary backbone that extracts features, and a fixed task-specific output layer that maps these features to the final prediction, as shown in Figure 1a. Given a mini-batch $\mathbf{x} = [\dots, \mathbf{x}^\mu, \dots]$ of $bs$ input samples $\mathbf{x}^\mu$, where $\mu \in \{1, \dots, bs\}$, we first obtain their binary representations $\mathbf{a}_0^\mu = bin(\mathbf{x}^\mu) \in \{\pm 1\}^{K_0}$. The binarization function can be realized via median thresholding for images or thermometer encoding for tabular data (Bacellar et al., 2024a). The resulting binary batch $\mathbf{a}_0$ is then fed to the binary backbone.

**Trainable BNN Backbone.** The backbone comprises a stack of $L$ fully-connected binary layers. For each layer $l \in \{1, \dots, L\}$, its state is defined by two matrices, following prior work on binary synapses (Baldassi, 2009). The first is the matrix of *hidden discrete weights* $\mathbf{H}_l \in \mathbb{Z}^{K_l \times K_{l-1}}$. These integer-valued weights encode the synaptic inertia of each connection, providing a mechanism for stable learning and mitigating catastrophic forgetting (Kirkpatrick et al., 2017). Although formulated as unbounded integers, in practice $\mathbf{H}_l$ is constrained to the finite range of a $B$-bit signed integer, $[-2^{B-1}, 2^{B-1} - 1]$. The second is the matrix of *visible binary weights* $\mathbf{W}_l = sign(\mathbf{H}_l) \in \{\pm 1\}^{K_l \times K_{l-1}}$, representing the effective weights used during the forward pass. Here, $K_l$ denotes the number of neurons in layer $l$, and the *sign* function is applied component-wise, returning $-1$ for negative inputs and $+1$ otherwise. For each layer $l$, the pre-activations and activations are computed as $\mathbf{z}_l = \mathbf{W}_l \mathbf{a}_{l-1}$ and $\mathbf{a}_l = sign(\mathbf{z}_l)$, respectively. The output of the BNN backbone (i.e., the activations $\mathbf{a}_L$) is then fed to the task-specific output layer to produce the final predictions $\hat{\mathbf{y}}$.

**Task-Specific Output Layer.** To make predictions, the binary features $\mathbf{a}_L^\mu$ of each sample $\mu$ must be mapped to the space of target variables $\mathbf{y}^\mu$. For regression tasks, this layer acts as a binary decoder $\{\pm 1\}^{K_L} \to \mathbb{R}^D$, where $D$ is the output dimensionality. For a $C$-class classification problem, the goal is to produce a logit vector $\hat{\mathbf{y}}^\mu \in \mathbb{R}^C$, whose largest component corresponds to the predicted class. While different mappings can be used, the choice of output layer influences both learning and accuracy. In this work, we use a linear layer represented by a fixed, randomly initialized matrix $\mathbf{P} \in \{\pm 1\}^{C \times K_L}$. This corresponds to associating a prototype vector $\boldsymbol{\rho}^c$ (the $c$-th row of $\mathbf{P}$) with each class $c \in \{1, \ldots, C\}$ (see Appendix C for the generation method). Providing the backbone with a stable and geometrically optimal set of target activations – the class prototypes – simplifies credit assignment during training and consistently yielded the best performance in our experiments. Although it is possible to train this classifier, we found empirically that keeping it fixed is more practical and effective. The final logits are computed as $\hat{\mathbf{y}} = \mathbf{P}\,\mathbf{a}_L$, and the predicted class for each sample $\mu$ is given by $\hat{c}^\mu = \arg\max_c \hat{\mathbf{y}}_c^\mu$.

## 3.2 Binary Error Backpropagation

The BEP learning rule is error-driven. A backward update is initiated for each sample $\mu$ when the logit $\hat{\mathbf{y}}_{c^\mu}^\mu$ associated with the ground-truth class $c^\mu$ is not sufficiently larger than the others. Specifically, an update is triggered if

$$\hat{\mathbf{y}}_{c^\mu}^\mu - \max_{c \neq c^\mu} \hat{\mathbf{y}}_c^\mu < rK_L, \tag{1}$$

where $r \in (0, 1]$ is a user-specified margin hyperparameter and $K_L$ is the size of the last hidden layer (also representing the maximum possible logit value). Higher values of $r$ enforce a larger gap between the correct-class logit and all others, thereby encouraging more robust classification.

For clarity, we describe the backward pass for a single training sample $\mu$, although in practice it is applied to all elements of a mini-batch. When a sample $\mu$ meets the update criterion in Eq. 1, BEP initiates the backward phase to adjust the hidden integer weights $\mathbf{H}_l$ so that the final output better aligns with the correct target. This is accomplished by defining, for each layer $l$, a binary *desired activation* vector $\mathbf{a}_l^{*\mu}$ that is propagated from the output layer $L$ back to the first layer.

**Desired Activation at the Last Layer.** The backward pass begins at the final layer $L$ of the BNN backbone. The first step is to determine the desired activation vector $\mathbf{a}_L^{*\mu}$, which is the ideal binary vector that maximizes the logit for the correct class $c^\mu$. Since the logit is a scalar product between activations and the class prototypes from the fixed classifier $\mathbf{P}$, the desired activations can be found analytically. They correspond to the prototype vector $\boldsymbol{\rho}^{c^\mu}$ for the correct class:

$$\mathbf{a}_L^{*\mu} := \arg\max_{\mathbf{a} \in \{\pm 1\}^{K_L}} \langle \mathbf{a}, \boldsymbol{\rho}^{c^\mu} \rangle = \boldsymbol{\rho}^{c^\mu}. \tag{2}$$

**Backpropagation of Desired Activations.** For each layer $l < L$, the desired activation vector $\mathbf{a}_l^{*\mu}$ is obtained by back-propagating the error signal $\mathbf{a}_{l+1}^{*\mu}$ from the subsequent layer $l+1$. The goal is to determine activations $\mathbf{a}_l^\mu$ that maximize alignment with desired activations $\mathbf{a}_{l+1}^{*\mu}$:

$$\arg\max_{\mathbf{a} \in \{\pm 1\}^{K_l}} \langle \mathbf{a}_{l+1}^{*\mu}, \, sign(\mathbf{W}_{l+1}\,\mathbf{a}) \rangle. \tag{3}$$

Since this is a combinatorial search over $2^{K_l}$ configurations, BEP solves a relaxed problem by removing the non-linear *sign* function and instead maximizing the alignment with the pre-activations:

$$\arg\max_{\mathbf{a} \in \{\pm 1\}^{K_l}} \langle \mathbf{a}_{l+1}^{*\mu}, \, \mathbf{W}_{l+1}\mathbf{a} \rangle. \tag{4}$$

This relaxed objective increases the magnitude of the pre-activations $\mathbf{z}_{l+1}$ while aligning their signs with $\mathbf{a}_{l+1}^{*\mu}$. A global optimum is unnecessary, as the goal is to steer weight updates in the appropriate direction. As shown in Lemma 1, the solution to this relaxed problem can be derived analytically.

**Backward Gating.** In addition, we introduce a gating mechanism to regulate the back-propagated signal. This gate is applied at layer $l+1$ to emphasize learning on neurons most amenable to change, i.e., those with pre-activations $\mathbf{z}_{l+1}^\mu = \mathbf{W}_{l+1}\mathbf{a}_l^\mu$ near the decision threshold of 0. Neurons with

large-magnitude pre-activations (nearly saturated responses) are excluded from the backward pass of sample $\mu$. This is realized through a binary gating vector $\mathbf{g}_{l+1}^{\mu}$:

$$(\mathbf{g}_{l+1}^{\mu})_i = \begin{cases} 1, & \text{if } |z_{l+1,i}^{\mu}| \leq \nu K_l \\ 0, & \text{otherwise} \end{cases} \quad (5)$$

where $z_{l+1,i}^{\mu}$ is the $i$-th pre-activation at layer $l + 1$ and $\nu \in [0, 1]$ is a tunable threshold. More formally, for a generic gating vector $\mathbf{g}$, we define the gated scalar product as $\langle \mathbf{a}, \mathbf{a}' \rangle_{\mathbf{g}} := \sum_i g_i a_i a_i'$. This modifies the relaxed optimization problem of Eq. 4 to:

$$\underset{\mathbf{a} \in \{\pm 1\}^{K_l}}{\arg\max} \left\langle \mathbf{a}_{l+1}^{*\mu}, \mathbf{W}_{l+1}\mathbf{a} \right\rangle_{\mathbf{g}_{l+1}^{\mu}} . \quad (6)$$

This gating function effectively filters the binary error signal, allowing it to pass only through neurons close to their activation boundary. This ensures that weight updates are driven by the parts of the BNN most susceptible to flipping their activations. The solution to Eq. 6 is given by the following lemma (proved in Appendix A).

**Lemma 1** (Desired activations). *Consider a binary vector $\mathbf{b} \in \{\pm 1\}^{K_b}$, a binary matrix $\mathbf{W} \in \{\pm 1\}^{K_b \times K_a}$, and a gating vector $\mathbf{g} \in \{0, 1\}^{K_b}$. Problem $\arg\max_{\mathbf{a} \in \{\pm 1\}^{K_a}} \langle \mathbf{b}, \mathbf{Wa} \rangle_{\mathbf{g}}$ has the unique solution:* $\mathbf{a}^* = sign(\mathbf{W}^\top (\mathbf{g} \odot \mathbf{b}))$.

Combining the base case from Eq. 2 with Lemma 1 yields the recursive expression for computing the desired activation vector at any layer $l$:

$$\mathbf{a}_l^{*\mu} = \begin{cases} \boldsymbol{\rho}^{c^{\mu}}, & \text{if } l = L \\ sign\left(\mathbf{W}_{l+1}^\top \left(\mathbf{g}_{l+1}^{\mu} \odot \mathbf{a}_{l+1}^{*\mu}\right)\right) & \text{if } l < L \end{cases}, \quad (7)$$

where $\odot$ denotes the element-wise product. This formulation forms the cornerstone of BEP, establishing a fully binary chain rule for propagating binary error signals throughout the BNN.

### 3.3 WEIGHT UPDATE MECHANISM

Once the target activation vector $\mathbf{a}_l^{*\mu}$ has been determined for layer $l$, the corresponding hidden weights $\mathbf{H}_l$ are updated. This update strengthens the association between the incoming activation vector $\mathbf{a}_{l-1}^{\mu}$ and the desired output pattern $\mathbf{a}_l^{*\mu}$. For every sample $\mu$ that triggers an update, we first compute a candidate weight-update matrix from the desired binary weights, following a procedure similar to that in prior work (Baldassi et al., 2007; Baldassi, 2009; Colombo et al., 2025). In particular, using the same strategy employed to obtain $\mathbf{a}_l^{*\mu}$, we maximize $\langle \mathbf{a}_l^{*\mu}, \mathbf{W}_l \mathbf{a}_{l-1}^{\mu} \rangle$ with respect to $\mathbf{W}_l$. The solution gives the desired direction of change for the integer weights[1]:

$$\Delta \mathbf{H}_l^{\mu} = sign\left(\mathbf{a}_l^{*\mu}(\mathbf{a}_{l-1}^{\mu})^\top\right) = \mathbf{a}_l^{*\mu}(\mathbf{a}_{l-1}^{\mu})^\top \in \{\pm 1\}^{K_l \times K_{l-1}}. \quad (8)$$

Eq. 8 is the natural matrix generalization of the classical supervised Hebbian perceptron rule (Rosenblatt, 1958) and of the Clipped Perceptron (CP) and CP+Reinforcement (CP+R) variants (Baldassi et al., 2007; Baldassi, 2009). When $K_l = 1$, the update $\Delta \mathbf{H}_l^{\mu}$ reduces exactly to the CP update, which adds (or subtracts) the input vector $\mathbf{a}_{l-1}^{\mu}$ to the synaptic stability variables whenever the desired output $\mathbf{a}_l^{*\mu}$ is $+1$ (or $-1$). In the multi-layer setting, Eq. 8 applies this outer-product mechanism independently to each hidden neuron, with the desired activations $\mathbf{a}_l^{*\mu}$ serving as binary targets.

This potential update is then filtered by a binary mask $\mathbf{M}_l^{\mu} \in \{0, 1\}^{K_l \times K_{l-1}}$ that selects which weights to modify. As shown in Appendix B, the resulting update is locally optimal with respect to the desired activations $\mathbf{a}_l^{*\mu}$. Following the sparse update strategy of (Colombo et al., 2025), the mask is constructed by partitioning the neurons of layer $l$ into subgroups and, within each group, updating only the incoming weights of the wrong perceptron deemed *easiest to correct* based on its pre-activation. This mechanism promotes a more uniform distribution of updates, leading to more stable training and improved generalization. If $\mathcal{M}$ denotes the set of mini-batch indices that trigger an update, we first compute each per-sample update $\Delta \mathbf{H}_l^{\mu}$ using the same pre-update hidden weights $\mathbf{H}_l$, and then apply the aggregated update as:

$$\mathbf{H}_l \leftarrow \mathbf{H}_l + 2 \sum_{\mu \in \mathcal{M}} (\mathbf{M}_l^{\mu} \odot \Delta \mathbf{H}_l^{\mu}). \quad (9)$$

---

[1]It follows from treating each row of $\mathbf{W}_l$ as an independent optimization and swapping $\mathbf{a}$ and $\mathbf{W}$ in Lemma 1.

The binary mask $\mathbf{M}_l^\mu$ implements a sparse *winner-takes-update* rule within each neuron group: only the least stable unit, i.e., the neuron with the smallest signed stability $\mathbf{a}_l^{*\mu} H_{l,j}$ among the misclassified ones, is updated. This mechanism bounds the number of synapses modified per pattern and prevents over-reinforcing neurons that already classify the sample with high confidence, effectively acting as a discrete, data-dependent form of learning-rate control. Additional analysis of mask density and its effect on convergence and generalization is provided in Appendix D.5. Finally, a reinforcement step from the CP+R rule stochastically strengthens existing memory trajectories of each integer weight $h \in \mathbf{H}_l$ via the update $h \leftarrow h + 2sign(h)$. This occurs with probability $p_r \sqrt{2/(\pi K_l)}$, where the reinforcement probability $p_r$ is rescaled each epoch by $\sqrt{E^e}$. This mechanism reinforces weights more frequently when the model is uncertain and balances its effect across layers of different sizes.

While (Colombo et al., 2025) employs a fixed group size $\gamma$, corresponding to a constant number of updates per epoch, BEP uses a scheduled group size. Let $\Gamma_l = \{d \in \mathbb{N} : d \mid K_l\}$ denote the ordered list of positive divisors of $K_l$. The algorithm begins with a user-defined initial group size $\gamma_{0,l} \in \Gamma_l$ and increases it adaptively based on accuracy. Whenever the generalization error stagnates for a user-defined number of epochs, the group size is increased to the next larger divisor $\gamma_{t+1,l} = \min \{d \in \Gamma_l : d > \gamma_{t,l}\}$. If $\gamma_{t,l}$ is already the largest divisor of $K_l$ (the last element of $\Gamma_l$), it remains fixed. Because only one perceptron per group is updated, enlarging the groups gradually reduces the number of weight changes at each step, resulting in increasingly sparse updates. Empirically, this mechanism leads to a more stable optimization process in the later stages of training.

### 3.4 Analogy to Gradient-Based Backpropagation

The BEP procedure can be viewed as a binary reformulation of the classical BP computational graph (Rumelhart et al., 1986). It preserves the global flow of information while substituting real-valued operations with binary, bit-wise counterparts. In standard BP, the weight update for layer $l$ follows a gradient-descent form:

$$\mathbf{W}_l \leftarrow \mathbf{W}_l - \eta\, \boldsymbol{\delta}_l\, \mathbf{a}_{l-1}^\top,$$

where $\eta$ is the learning rate and $\boldsymbol{\delta}_l$ denotes the back-propagated error signal for layer $l$. This real-valued signal is computed recursively by the chain rule:

$$\boldsymbol{\delta}_l = \begin{cases} \nabla_{\mathbf{a}_L}\mathcal{L} \odot \sigma'(\mathbf{z}_L), & \text{if } l = L \\ (\mathbf{W}_{l+1}^\top \boldsymbol{\delta}_{l+1}) \odot \sigma'(\mathbf{z}_l), & \text{if } l < L \end{cases}. \tag{10}$$

Each component of BEP mirrors a counterpart in gradient-based BP. The binary desired activation $\mathbf{a}_l^{*\mu}$ serves as the binary analog of the error signal. Back-projecting it through $\mathbf{W}_{l+1}^\top$ is structurally equivalent to the error propagation step in the continuous case. The binary gating function plays the role of the activation derivative $\sigma'(\cdot)$, blocking error transmission through saturated neurons as $\sigma'(\cdot)$ approaches zero for large inputs. Finally, the sparse mask $\mathbf{M}_l$ fulfills the role of the learning rate: whereas $\eta$ controls the magnitude of an update, the mask governs its sparsity. These correspondences show that BEP constitutes a coherent end-to-end binary analog of traditional gradient-based BP.

### 3.5 Applying BEP-TT to Recurrent Architectures

One of the main strengths of BEP is that its global error-propagation strategy directly extends to RNNs, a class of models that require explicit temporal credit assignment and is generally intractable for local learning rules. By unrolling the NN through time, we obtain a binary counterpart of the BP Through Time (BPTT) algorithm, which we call BEP-TT. Demonstrating its effectiveness provides key validation of BEP, showing that a global, end-to-end binary error signal can train recurrent architectures. Accordingly, we employ a binary RNN for many-to-one sequence classification, as shown in Figure 1b. Given an input sequence $\mathbf{x}^\mu = (\mathbf{x}_1^\mu, \ldots, \mathbf{x}_T^\mu)$ of length $T$, the RNN is defined by state-to-state weights $\mathbf{H}_{ss} \in \mathbb{Z}^{K_s \times K_s}$, input-to-state weights $\mathbf{H}_{xs} \in \mathbb{Z}^{K_s \times K_x}$, and state-to-output weights $\mathbf{H}_{sy} \in \mathbb{Z}^{K_y \times K_s}$, along with their binary counterparts $\mathbf{W}_{ss}$, $\mathbf{W}_{xs}$, and $\mathbf{W}_{sy}$.

**Forward Pass Through Time.** At each time step $t \in 1, \ldots, T$, the state vector $\mathbf{s}_t^\mu \in \{\pm1\}^{K_s}$ is computed from the previous state $\mathbf{s}_{t-1}^\mu$ and the current binarized input $\mathbf{a}_t^\mu \in \{\pm1\}^{K_x}$ according to the recurrence $\mathbf{s}_t^\mu = sign(\mathbf{W}_{xs}\mathbf{a}_t^\mu + \mathbf{W}_{ss}\mathbf{s}_{t-1}^\mu)$. After the final time step $T$, the state $\mathbf{s}_T^\mu$ is mapped to an output activation $\mathbf{s}_y^\mu \in \{\pm1\}^{K_y}$ via $\mathbf{s}_y^\mu = sign(\mathbf{W}_{sy}\mathbf{s}_T^\mu)$. This activation is then passed to the fixed classifier $\mathbf{P}$ to produce the prediction $\hat{\mathbf{y}}^\mu$, which is used to check the update criterion from Eq. 1.

**Binary Backpropagation Through Time.** When a sequence $\mathbf{x}^\mu$ satisfies the update criterion in Eq. 1, the BEP-TT procedure is performed. This mechanism parallels BEP's operation on feedforward NNs but operates over the unrolled temporal structure, propagating a target state vector $\mathbf{s}_t^{*\mu}$ backward from time step $t = T$ to $t = 1$. First, the desired output state is set to the correct class prototype, $\mathbf{s}_y^{*\mu} := \boldsymbol{\rho}^{c^\mu}$, which provides the error signal for the state-to-output layer. Using Eq. 7, this signal is then back-projected to obtain the desired state at the last time step, $\mathbf{s}_T^{*\mu} = sign\left(\mathbf{W}_{sy}^\top\left(\mathbf{g}_y^\mu \odot \mathbf{s}_y^{*\mu}\right)\right)$. For preceding time steps $t \in \{T - 1, \ldots, 1\}$, the desired state is propagated recursively:

$$\mathbf{s}_t^{*\mu} = sign\left(\mathbf{W}_{ss}^\top\left(\mathbf{g}_{t+1}^\mu \odot \mathbf{s}_{t+1}^{*\mu}\right)\right).$$

Here, $\mathbf{g}_y^\mu$ and $\mathbf{g}_{t+1}^\mu$ are the backward gates computed from the pre-activations $\mathbf{z}_y^\mu$ and $\mathbf{z}_{t+1}^\mu$. This recursion propagates the binary error signal through the sequence, enabling temporal credit assignment.

**Accumulated Updates for Shared Weights.** Because RNNs share weights across time, the matrices $\mathbf{H}_{xs}$ and $\mathbf{H}_{ss}$ must accumulate updates over all time steps for all samples $\mu \in \mathcal{M}$ that triggered an update. These aggregated updates are then applied to the integer-valued hidden weights using the corresponding binary masks $\mathbf{M}_{xs}^\mu$ and $\mathbf{M}_{ss}^\mu$, as described in Eq. 8 and Eq. 9:

$$\mathbf{H}_{xs} \leftarrow \mathbf{H}_{xs} + 2\sum_{\mu \in \mathcal{M}} \mathbf{M}_{xs}^\mu \odot \sum_{t=1}^T \left(\mathbf{s}_t^{*\mu}(\mathbf{a}_t^\mu)^\top\right), \quad \mathbf{H}_{ss} \leftarrow \mathbf{H}_{ss} + 2\sum_{\mu \in \mathcal{M}} \mathbf{M}_{ss}^\mu \odot \sum_{t=1}^T \left(\mathbf{s}_t^{*\mu}(\mathbf{s}_{t-1}^\mu)^\top\right).$$

These updates are followed by the reinforcement step described in the previous section. The binary masks $\mathbf{M}_{xs}^\mu$ and $\mathbf{M}_{ss}^\mu$ are shared across time steps to preserve weight tying, ensuring masking commutes with temporal aggregation. Allowing masks to vary with time would violate this property, effectively training different copies of the same parameter at each time index. The final state-to-output layer is updated analogously to the feedforward case described in Section 3.2:

$$\mathbf{H}_{sy} \leftarrow \mathbf{H}_{sy} + \sum_{\mu \in \mathcal{M}} \left(\mathbf{M}_{sy}^\mu \odot \mathbf{s}_y^{*\mu}(\mathbf{s}_T^\mu)^\top\right).$$

This BEP-TT formulation yields a fully binary training procedure for binary RNNs. Its ability to perform temporal credit assignment stems from the global error propagation mechanism of BEP, a key advantage over local learning rules.

## 4 EXPERIMENTAL EVALUATION

In this section, we empirically evaluate the proposed BEP algorithm. The experiments have two main objectives: *(i)* to benchmark BEP against the SotA method for binary training of MLPs (Colombo et al., 2025), and *(ii)* to demonstrate the generality of BEP's global credit assignment by applying it to binary RNNs, a setting for which local learning methods are unsuited. The Python code used in our experiments is available in a public repository[2]. Section 4.1 describes the experimental setup. Section 4.2 evaluates BEP on binary MLPs against both the SotA approach and QAT-based methods that rely on full-precision BP. Section 4.3 tests BEP on binary RNNs, contrasting its performance with QAT. Finally, Section 4.4 explores the effect of the gating hyperparameter $\nu$ on training dynamics.

### 4.1 EXPERIMENTAL SETUP

Our evaluation spans a diverse set of benchmarks. For the feedforward MLPs, we adopt the same datasets as (Colombo et al., 2025): Random Prototypes, FashionMNIST (Xiao et al., 2017), CIFAR-10 (Krizhevsky et al., 2009), and Imagenette (Howard, 2019). The Random Prototypes dataset comprises 20,000 training and 3,000 test samples with input dimension $K_0 = 1000$ and a flip probability $p = 0.46$. FashionMNIST contains 50,000 training and 10,000 test grayscale images of size $28 \times 28$. For CIFAR-10 and Imagenette, we use the same AlexNet-derived features as in (Colombo et al., 2025). Both are 10-class datasets; CIFAR-10 has 50,000 training and 10,000 test samples, while Imagenette has 10,000 training and 1,963 test samples. To evaluate recurrent models, we use Sequential MNIST (S-MNIST) (Deng, 2012) and 30 real-world time-series classification tasks from the UCR Archive (Dau et al., 2019), which cover a wide range of sample sizes, feature dimensions, and class counts. In all experiments, we utilize a fixed output classifier $\mathbf{P}$ generated via the binary equiangular frame method detailed in Appendix C.

---

[2]https://github.com/AI-Tech-Research-Lab/BEP

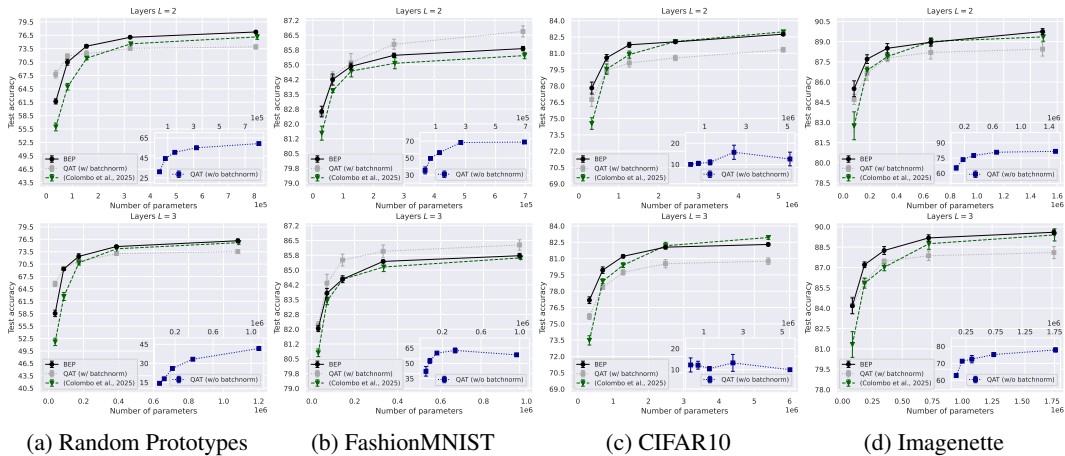

(a) Random Prototypes     (b) FashionMNIST     (c) CIFAR10     (d) Imagenette

Figure 2: Test accuracy as a function of the number of parameters on Random Prototypes, Fashion-MNIST, CIFAR10, and Imagenette. Results compare BEP with both the SotA approach (Colombo et al., 2025) and QAT-based methods for binary MLPs with $L = 2$ and $L = 3$ hidden layers.

## 4.2 PERFORMANCE ON BINARY MULTI-LAYER PERCEPTRONS

Our first experiment evaluates BEP against the current SotA approach for binary training of MLPs (Colombo et al., 2025). Specifically, we considered binary MLPs with two and three hidden layers, trained using both BEP and the SotA local learning rule. The tuned hyperparameters, across $e = 50$ epochs, include the robustness parameter $r$, reinforcement probability $p_r$, initial group size $\gamma_{0,l}$, and gating threshold $\nu$. For a comprehensive benchmark, we also trained the same architectures using the QAT implementation from the *Larq* framework (Geiger & Team, 2020). In this setup, the forward pass employs binarized weights, while the backward pass updates latent FP parameters with the Adam optimizer (Kingma & Ba, 2014). To ensure a fair comparison, the main QAT baseline does not use batch normalization, resulting in a fully binary model at inference. For completeness, we also report the performance of QAT with batch normalization as a reference point.

Figure 2 reports test accuracy, averaged over 5 runs, on four datasets as a function of the parameter count. The results show that both fully binary approaches – BEP and the SotA local rule – significantly outperform the comparable QAT baseline across all configurations. When comparing binary methods, BEP consistently surpasses the SotA approach, achieving improvements of up to $+6.89\%$, $+1.22\%$, $+3.70\%$, and $+2.85\%$ on Random Prototypes, FashionMNIST, CIFAR-10, and Imagenette, respectively, at the smallest parameter configurations. As model size increases, the performance gap narrows, with BEP matching the local credit assignment rule but falling slightly behind in one high-parameter setting (CIFAR-10, 3 layers). These findings underscore the importance of global error propagation for effective credit assignment in binary MLPs, while also showing that BEP outperforms both local learning rules and standard QAT baselines without relying on FP gradients.

## 4.3 VALIDATION ON BINARY RECURRENT NEURAL NETWORKS

Our second experiment tests the BEP algorithm on binary RNNs using its time-unrolled variant BEP-TT described in Section 3.5. The goal is to demonstrate that a global, end-to-end error signal can effectively train recurrent models, a task typically intractable for purely local learning rules. Specifically, we evaluate BEP-TT on many-to-one sequence classification tasks across 30 datasets from the UCR Time Series Archive (Dau et al., 2019), using only the last window of each time series. All RNN results use 3-fold cross-validation and 3 independent runs with the same hyperparameters: robustness $r = 0.5$, reinforcement probability $p_r = 0.5$, initial group size $\gamma_{0,l} = 15$, gating threshold $\nu = 0.05$, hidden and output layer sizes $K_s = K_y = 1035$, training epochs $e = 50$, and batch size $bs = N/10$. As a baseline, we trained binary RNNs with the same architecture using the QAT implementation from the *Larq* framework (Geiger & Team, 2020). As in the previous experiment, no batch or layer normalization was applied so that the models remain fully binary at inference time. For completeness, we report the performance of QAT with batch normalization as a reference point.

Table 1: Test accuracy on 30 UCR datasets. The results compare our BEP algorithm with the QAT-based training from the *Larq* framework (Geiger & Team, 2020) for binary RNN architectures.

| UCR Dataset | BEP | QAT w/o batchnorm | QAT w/ batchnorm | UCR Dataset | BEP | QAT w/o batchnorm | QAT w/ batchnorm |
|---|---|---|---|---|---|---|---|
| *ArticularyWordRec.* | **81.28 ± 2.99** | 51.94 ± 3.61 | 77.62 ± 3.80 | *JapaneseVowels* | **95.47 ± 1.24** | 84.06 ± 2.92 | 96.25 ± 1.01 |
| *Cricket* | **86.85 ± 4.19** | 61.30 ± 6.88 | 86.11 ± 6.03 | *MelbournePedestrian* | **73.03 ± 4.59** | 42.83 ± 2.35 | 90.91 ± 0.90 |
| *DistalPOAgeGroup* | **79.78 ± 2.65** | 73.84 ± 2.51 | 76.94 ± 3.27 | *MoteStrain* | **78.62 ± 2.29** | 74.42 ± 1.70 | 79.82 ± 1.81 |
| *ECG5000* | **91.40 ± 0.53** | 88.20 ± 1.08 | 92.71 ± 0.74 | *MotionSenseHAR* | **74.25 ± 2.07** | 67.97 ± 1.80 | 77.36 ± 1.89 |
| *ECGFiveDays* | **81.79 ± 1.97** | 69.42 ± 2.71 | 95.25 ± 1.45 | *PEMS-SF* | **86.13 ± 3.66** | 60.69 ± 4.55 | 85.83 ± 2.43 |
| *ERing* | **82.44 ± 4.90** | 71.67 ± 3.89 | 79.11 ± 4.93 | *PenDigits* | **97.13 ± 0.21** | 66.99 ± 1.44 | 99.00 ± 0.21 |
| *Earthquakes* | **80.34 ± 2.94** | 79.83 ± 2.83 | 77.01 ± 2.96 | *ProximalPOAgeGroup* | **82.64 ± 3.08** | 77.24 ± 3.64 | 82.20 ± 2.27 |
| *Epilepsy2* | **92.45 ± 0.53** | 88.82 ± 0.58 | 93.72 ± 0.42 | *ProximalPOCorrect* | **78.26 ± 1.09** | 72.76 ± 2.89 | 75.68 ± 2.66 |
| *FreezerRegularTrain* | **78.21 ± 1.17** | 75.26 ± 1.56 | 85.92 ± 1.06 | *ProximalPhalanxTW* | **79.83 ± 2.66** | 74.26 ± 5.11 | 75.59 ± 3.03 |
| *FreezerSmallTrain* | **78.16 ± 1.46** | 75.02 ± 1.44 | 85.60 ± 1.50 | *SmoothSubspace* | **89.00 ± 4.24** | 58.00 ± 4.83 | 90.33 ± 4.08 |
| *GunPtAgeSpan* | **86.18 ± 2.27** | 81.08 ± 4.53 | 84.11 ± 2.55 | *SonyAIBORobotSurf1* | **75.68 ± 2.73** | 64.14 ± 2.05 | 80.68 ± 3.04 |
| *GunPtOldVersusYoung* | **94.01 ± 1.57** | 91.87 ± 2.91 | 93.79 ± 1.26 | *SonyAIBORobotSurf2* | **82.04 ± 2.37** | 73.10 ± 1.71 | 84.59 ± 2.62 |
| *InsectEPGRegularTrain* | **99.14 ± 0.71** | 94.64 ± 2.97 | 99.04 ± 0.64 | *StarLightCurves* | **82.32 ± 0.34** | 81.96 ± 0.49 | 82.68 ± 0.46 |
| *InsectEPGSmallTrain* | **99.37 ± 0.56** | 96.62 ± 2.43 | 98.75 ± 0.83 | *Tiselac* | **81.63 ± 0.47** | 64.52 ± 0.98 | 84.81 ± 0.25 |
| *ItalyPowerDemand* | **94.65 ± 1.22** | 83.18 ± 2.01 | 96.23 ± 1.01 | *Wafer* | **95.92 ± 0.61** | 95.25 ± 0.64 | 97.93 ± 0.24 |

For all datasets and models, inputs were binarized with a distributive thermometer encoder (Bacellar et al., 2024a), followed by a fixed $\pm 1$ expansion layer projecting the dimension to $K_0 = 1035$. A hyperparameter search was conducted to tune the sequence window length, defined as the number of timesteps included in the temporal window, and the number of thermometer bits.

Table 1 presents the test accuracy across the considered UCR tasks. On every dataset, BEP-TT consistently outperforms the comparable QAT baseline in training fully binary RNNs, achieving an average test accuracy improvement of $+10.57\%$. These results validate the proposed binary error propagation mechanism beyond feedforward MLPs. Moreover, because BEP relies on bitwise operations even during training, it substantially reduces memory and computational costs compared to QAT, which requires full-precision Adam updates, as discussed in Section 4.5.

## 4.4 THE ROLE OF THE GATING THRESHOLD $\nu$

A crucial element to the generalization performance of the BEP algorithm is its gating mechanism, which modulates the backward error signal as described in Section 3.2. In this third experiment, we examine how varying the threshold $\nu$ affects the validation accuracy of a binary RNN on the S-MNIST dataset. The following hyperparameters are used: robustness $r = 0.5$, reinforcement probability $p_r = 0.5$, initial group size $\gamma_{0,l} = 15$, layer sizes $K_s = K_y = 1875$, training epochs $e = 50$, and batch size $bs = 100$. Corresponding results for binary MLPs are provided in Appendix D.1.

Figures 3a and 3b, which show the accuracy averaged over 3 runs as a function of window length and gating threshold $\nu$, respectively, highlight the crucial role of this mechanism. Its effect becomes more pronounced as the window length grows (corresponding to deeper backward steps through time), emphasizing the importance of focusing updates on neurons whose pre-activations lie near the decision boundary. The results reveal an optimum in $\nu$: very low or very high thresholds degrade performance, whereas intermediate values (around $10^{-2}$) consistently yield the highest accuracy across different temporal depths. By filtering out saturated neurons from the error signal, the gate ensures that weight updates focus on parts of the BNN most susceptible to flipping their activations.

## 4.5 DISCUSSION AND LIMITATIONS

A key advantage of BEP lies in its ability to perform training using only bitwise computations. Table 2 summarizes the memory footprint and computational complexity of BEP compared with standard QAT. Relative to QAT-based methods, our approach achieves a $2\times$ reduction in memory usage for hidden weights and a $32\times$ reduction for error signals and weight updates. Moreover, Adam-based QAT requires an additional $64$ bits per parameter to store its first- and second-order moment estimates, further increasing its memory demand. From a computational standpoint, we adopt the hardware-level metric of equivalent Boolean gates (Colombo et al., 2025) to compare the intrinsic cost of arithmetic operations. All operations in BEP can be implemented using primitives such as XNOR, Popcount, and increment/decrement operations, which require at most $\mathcal{O}(10N)$ Boolean gates for $N$-bit operands (Bacellar et al., 2024b). In contrast, IEEE-754 single-precision

Table 2: Comparison of memory footprint and computational complexity between QAT (with Adam) and BEP. Memory is reported in bits per element, and complexity denotes the order of magnitude of equivalent Boolean-gate operations per element (Colombo et al., 2025). $^{\dagger}$: This estimate assumes the absence of batch normalization and full-precision scaling factors commonly used in QAT.

| Method | Memory (Bits) | | | Complexity (Boolean gates) | | |
|---|---|---|---|---|---|---|
| | Weights | Activations | Errors / Gradients | Forward | Backward | Update |
| QAT (Adam) | 32 (FP32) | 1 | 32 + 64 (Moments) | $\sim 10^{\dagger}$ | $\sim 10^4$ | $\sim 10^4$ |
| **BEP** | **16 (Int16)** | **1** | **1** | $\mathbf{\sim 10}$ | $\mathbf{\sim 10}$ | $\mathbf{\sim 10}$ |

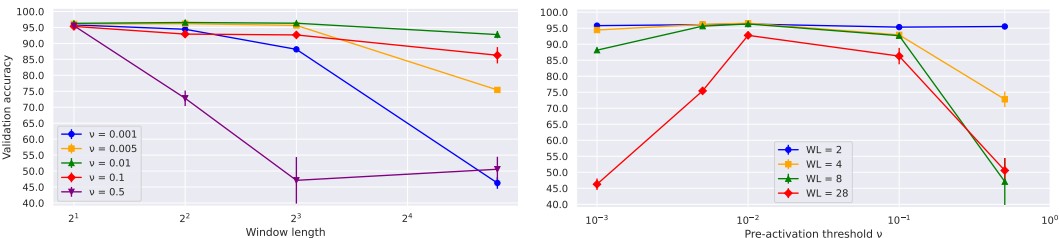

(a) Accuracy as a function of window length      (b) Accuracy as a function of gating threshold $\nu$

Figure 3: Validation accuracy of a binary RNN trained with BEP on the S-MNIST dataset for different values of the gating threshold hyperparameter $\nu$ and window length WL.

FP additions and multiplications (876, 2019), which underpin Adam-based QAT, are estimated to require on the order of $10^4$ Boolean gates each (Luo & Sun, 2024). This comparison highlights the substantial computational advantage of BEP, which reduces the Boolean-gate cost by approximately three orders of magnitude relative to FP32 QAT with Adam.

Despite its promising results, our approach has some limitations that suggest natural directions for future research. First, we focus primarily on binary MLPs and RNNs. Extending BEP to convolutional or transformer-style models requires a full binary design for these architectures, including handling binary convolutions, filter-level masking, and additional adaptations to support weight sharing, spatial structure, and multi-head mechanisms. Second, our experiments are restricted to classification tasks. Although BEP naturally supports arbitrary binary output vectors and could, in principle, be applied to multi-label prediction, binary segmentation, or more general binary-vector regression, such tasks necessitate additional design choices regarding the output encoding. Third, we evaluate BEP on mid-scale datasets and moderate-depth NNs. Extending BEP to large-scale models (e.g., ImageNet-level CNNs (Bulat & Tzimiropoulos, 2019; Vargas et al., 2024)) requires substantial architectural adaptations to convolutional or transformer backbones. Moreover, binary-compatible normalization mechanisms may become necessary to ensure stable training in such settings.

## 5    CONCLUSION AND FUTURE WORK

In this paper, we introduced BEP, an algorithm for training multi-layer BNNs using exclusively binary computations. The central contribution is the formulation of a principled, binary analog of the BP algorithm. By defining a recursive rule for propagating binary-valued error signals and updating integer-valued metaplastic weights, BEP bridges the gap between the global credit assignment of gradient-based learning and the computational efficiency of bitwise operations. Theoretically, this work shows that effective end-to-end learning in multi-layer binary NNs is possible without relying on continuous gradients, opening new avenues for analyzing discrete optimization in DL. Practically, BEP enables efficient training using only XNOR, Popcount, and increment/decrement operations, making it well-suited for constrained settings such as TinyML (Capogrosso et al., 2024; Pavan et al., 2024), privacy-preserving DL with homomorphic encryption (Falcetta & Roveri, 2022; Colombo et al., 2024), and neuromorphic systems (Indiveri & Liu, 2015; Yamazaki et al., 2022). Future work includes extending BEP to convolutional architectures, refining adaptive strategies for the gating threshold $\nu$, exploring learnable masking mechanisms, and developing a formal convergence analysis.

ACKNOWLEDGEMENTS

This paper is supported by Dhiria S.r.l. and by PNRR-PE-AI FAIR project funded by the NextGeneration EU program.

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

# A ANALYSIS OF THE BACKWARD PASS

It is possible to provide a justification for the core backward propagation rule. In particular, we show that the recursion for the desired activations is the *exact* optimizer of a tractable linear surrogate of the otherwise intractable combinatorial credit-assignment problem. This positions BEP as a principled binary analog of BP rather than an ad-hoc heuristic. At layer $l$, one would ideally choose the binary activation vector $\mathbf{a}_l^{*\mu}$ that, when mapped through $\mathbf{W}_{l+1}$ and binarized, maximizes alignment with the upper-layer target $\mathbf{a}_{l+1}^{*\mu}$. From Eq. 3:

$$\underset{\mathbf{a} \in \{\pm 1\}^{K_l}}{\arg\max} \langle \mathbf{a}_{l+1}^{*\mu}, \; sign(\mathbf{W}_{l+1}\mathbf{a}) \rangle.$$

This is a nonconvex combinatorial optimization over the hypercube with a discontinuous objective. In general, it is NP-hard by reduction from standard binary optimization problems. Therefore, we do not attempt to solve it exactly. Instead, we consider the linear surrogate obtained by dropping the nonlinearity inside the inner product. From Eq. 4:

$$\underset{\mathbf{a} \in \{\pm 1\}^{K_l}}{\arg\max} \langle \mathbf{a}_{l+1}^{*\mu}, \mathbf{W}_{l+1}\mathbf{a} \rangle.$$

Equivalently, the set may be relaxed to the hypercube $[-1, 1]^{K_l}$ and the optimum remains at a vertex.

**Proposition 1** (BEP back-projection solves the linear surrogate exactly). *Let $\mathbf{v} := \mathbf{W}_{l+1}^\top \mathbf{a}_{l+1}^{*\mu} \in \mathbb{R}^{K_l}$. The set of maximizers of Eq. 4 is*

$$\mathbf{a}_l^{*\mu} \in sign(\mathbf{v}) := \{\mathbf{a} \in \{\pm 1\}^{K_l} : a_i = sign(v_i) \; if \; v_i \neq 0, \; a_i \in \{\pm 1\} \; if \; v_i = 0\}, \quad (11)$$

*i.e., any coordinate-wise sign choice consistent with $\mathbf{v}$. In particular, when no coordinate tie occurs, $\mathbf{a}_l^{*\mu} = sign(\mathbf{W}_{l+1}^\top \mathbf{a}_{l+1}^{*\mu})$, which is exactly the BEP recursion (Eq. 7 without gating).*

*Proof.* By the adjoint identity $\langle \mathbf{u}, \mathbf{A}\mathbf{v} \rangle = \langle \mathbf{A}^\top \mathbf{u}, \mathbf{v} \rangle$,

$$\langle \mathbf{a}_{l+1}^{*\mu}, \; \mathbf{W}_{l+1}\mathbf{a} \rangle = \langle \mathbf{W}_{l+1}^\top \mathbf{a}_{l+1}^{*\mu}, \; \mathbf{a} \rangle = \sum_{i=1}^{K_l} v_i a_i.$$

The objective is separable across coordinates on the product set $\{\pm 1\}^{K_l}$, so it is maximized by choosing each $a_i$ to maximize $v_i a_i$, i.e., $a_i = sign(v_i)$ if $v_i \neq 0$ and any $a_i \in \{\pm 1\}$ if $v_i = 0$. $\quad \square$

**Lemma 2** (Convex relaxation has an integral optimum). *The convex relaxation of Eq. 4 with $\mathbf{a} \in [-1, 1]^{K_l}$ has the same optimal value, and the set of maximizers is*

$$\left\{ \mathbf{a} \in [-1, 1]^{K_l} : a_i = clip\left(\frac{v_i}{|v_i|}\right) \right\},$$

*which reduces to Eq. 11 at the vertices. Hence the linear surrogate is solved exactly at a binary point.*

*Proof.* Maximizing a linear function over a hypercube attains the optimum at a vertex. Coordinate-wise, the same separability argument as above applies. $\quad \square$

**Including the gate.** Recall the binary gate from Eq. 5 and write $\mathbf{D}_{l+1}^\mu := diag(\mathbf{g}_{l+1}^\mu)$. The BEP recursion with gating from Eq. 7 replaces $\mathbf{a}_{l+1}^{*\mu}$ by $\mathbf{D}_{l+1}^\mu \mathbf{a}_{l+1}^{*\mu}$, i.e., it solves the *masked* surrogate

$$\underset{\mathbf{a} \in \{\pm 1\}^{K_l}}{\arg\max} \langle \mathbf{D}_{l+1}^\mu \mathbf{a}_{l+1}^{*\mu}, \mathbf{W}_{l+1}\mathbf{a} \rangle \quad \Longrightarrow \quad \mathbf{a}_l^{*\mu} \in sign(\mathbf{W}_{l+1}^\top \mathbf{D}_{l+1}^\mu \mathbf{a}_{l+1}^{*\mu}). \quad (12)$$

Thus, the gate simply zeros out saturated coordinates of the upper-layer target before back-projection, directly mirroring the role of derivative clipping in STE-based BP.

Proposition 1 and Lemma 2 show that the BEP backward rule is the analytical optimizer of a well-posed linear objective approximating the intractable target-selection in Eq. 3. The gate induces a diagonal mask in that objective, yielding the exact masked optimizer in Eq. 12. Practically, this explains why BEP focuses the learning signal on neurons near their decision boundary (unsaturated coordinates) and provides a principled binary analog of gradient gating used by STE-based methods.

## A.1 Proof of Lemma 1

*Proof.* The gated scalar product can be expressed as $\langle \mathbf{b}, \mathbf{b}' \rangle_{\mathbf{g}} = \langle \mathbf{g} \odot \mathbf{b}, \mathbf{b}' \rangle$ and by the adjoint identity $\langle \mathbf{b}, \mathbf{Wa} \rangle = \langle \mathbf{W}^\top \mathbf{b}, \mathbf{a} \rangle$. Combining these two relations leads to

$$\langle \mathbf{b}, \mathbf{Wa} \rangle_{\mathbf{g}} = \langle \mathbf{W}^\top (\mathbf{g} \odot \mathbf{b}), \mathbf{a} \rangle.$$

Denoting $\mathbf{W}^\top (\mathbf{g} \odot \mathbf{b})$ as $\mathbf{z}$ of components in $\mathbb{Z} \setminus \{0\}$, we reduce to problem $\arg\max_{\mathbf{a}} \langle \mathbf{z}, \mathbf{a} \rangle$. The objective is separable across coordinates: $\langle \mathbf{z}, \mathbf{a} \rangle = \sum_i z_i a_i$, so it is maximized by choosing each $a_i$ to maximize $z_i a_i$, i.e., $a_i = sign(z_i)$. $\qquad \square$

## B  Local Correctness of the Weight Update

Beyond justifying the backward pass, we also show that the resulting weight update is beneficial in a layer–local sense. The next lemma proves that, for any sample triggering an update, the modification to the hidden integers $\mathbf{H}_l$ is guaranteed to be corrective: it pushes the neuron *stabilities* in the direction of the desired activation and increases an anchored alignment potential by a fixed, known amount.

**Lemma 3** (Local update correctness on the stabilities). *Fix a sample $\mu \in \mathcal{M}$ that triggers an update. Let $l$ be a layer and $j$ a neuron selected by the neuron-wise mask (i.e., the $j$-th column). Denote the hidden weights before and after the update by $\mathbf{H}_l$ and $\mathbf{H}'_l$, respectively, and define the stabilities*

$$u^\mu_{l,j} := \langle \mathbf{a}^\mu_{l-1}, \mathbf{h}_{l,j} \rangle, \qquad u'^\mu_{l,j} := \langle \mathbf{a}^\mu_{l-1}, \mathbf{h}'_{l,j} \rangle.$$

*If the neuron-wise update is $\Delta \mathbf{h}^\mu_{l,j} = 2a^{*\mu}_{l,j} \mathbf{a}^\mu_{l-1}$ (and zero otherwise), then the alignment strictly increases by a fixed amount:*

$$a^{*\mu}_{l,j} u'^\mu_{l,j} = a^{*\mu}_{l,j} u^\mu_{l,j} + 2K_{l-1} > a^{*\mu}_{l,j} u^\mu_{l,j}.$$

*Proof.*

$$u'^\mu_{l,j} = \langle \mathbf{a}^\mu_{l-1}, \mathbf{h}_{l,j} + 2a^{*\mu}_{l,j} \mathbf{a}^\mu_{l-1} \rangle = u^\mu_{l,j} + 2a^{*\mu}_{l,j} \|\mathbf{a}^\mu_{l-1}\|^2_2.$$

Since $\mathbf{a}^\mu_{l-1} \in \{\pm 1\}^{K_{l-1}}$, $\|\mathbf{a}^\mu_{l-1}\|^2_2 = K_{l-1}$. Multiplying by $a^{*\mu}_{l,j} \in \{\pm 1\}$ yields the claim. $\qquad \square$

**Remark 1** (From stability to visible pre-activation). *The forward pre-activations use visible weights $\mathbf{W}_l = sign(\mathbf{H}_l)$, hence $z^\mu_{l,j} = \langle \mathbf{a}^\mu_{l-1}, \mathbf{w}_{l,j} \rangle$ can change discontinuously when entries of $\mathbf{h}_{l,j}$ cross zero. Nevertheless, Lemma 3 implies monotonic drift of each coordinate of $\mathbf{h}_{l,j}$ toward the signed target $a^{*\mu}_{l,j} \mathbf{a}^\mu_{l-1}$. After $T$ updates of neuron $j$, each entry has shifted by $2T$ in the correct direction. Consequently, once*

$$T \geq \max_i \left\lceil \frac{|\mathbf{H}_{l,ij}(0)| + 1}{2} \right\rceil,$$

*all entries align with $a^{*\mu}_{l,j} \mathbf{a}^\mu_{l-1,i}$, and the visible pre-activations satisfy $sign(z^\mu_{l,j}) = a^{*\mu}_{l,j}$ and remain stable under further updates on the anchored desired activations.*

Lemma 3 shows that each neuron update yields a *strict quantifiable* increase of an anchored alignment by $2K_{l-1}$ (a discrete analog of a guaranteed descent step). Together with Remark 1, this ensures that repeated anchored updates drive the visible state toward the desired activation and stabilize it once sufficient integer margin accumulates. A full convergence proof is left for future work.

## C  Generating a Fixed Binary Classifier via Equiangular Frames

As stated in Section 3, our empirical results show that BEP achieves its best performance when using a fixed output classifier $\mathbf{P}$ whose class prototypes are geometrically well-separated. This approach is inspired by the concept of Equiangular Tight Frames (ETFs), which have been shown to emerge in the final layers of deep NNs during a phenomenon known as neural collapse (Papyan et al., 2020). While a simple, randomly generated classifier offers a baseline, optimizing the structure of these prototypes significantly improves class separability. This appendix details our method for generating a structured binary classifier by constructing a set of prototype vectors that are maximally and uniformly distant from each other in the binary feature space.

Neural collapse describes an empirical phenomenon where, in the terminal phase of training, the last-layer feature representations for all samples of a given class collapse to a single point (their class mean). Furthermore, the set of these class-mean vectors, along with the final-layer classifier weights, form a simplex ETF. A simplex ETF is a geometric configuration of vectors that are maximally separated from one another, characterized by equal norms and a constant, negative pairwise inner product. This structure is optimal for linear classification.

The classical ETF is defined in a real-valued vector space $\mathbb{R}^D$. In BNNs, we are interested in a binary analog where the feature vectors and classifier weights lie on the hypercube $\{\pm 1\}^D$. We define a *Binary Equiangular Frame* (BEF) as a set of $C$ binary vectors $\{\boldsymbol{\rho}_1, \ldots, \boldsymbol{\rho}_C\}$, where $\boldsymbol{\rho}_c \in \{\pm 1\}^D$, that satisfy two properties. The first property is high pairwise separation, i.e. the inner product $\langle \boldsymbol{\rho}_i, \boldsymbol{\rho}_j \rangle$ for $i \neq j$ should be as small (i.e., as negative) as possible. In the binary domain, this is equivalent to maximizing the Hamming distance between any two vectors. The second property is equiangularity, i.e. the inner products for all distinct pairs $\langle \boldsymbol{\rho}_i, \boldsymbol{\rho}_j \rangle$ should be approximately equal. Such a frame, when used as the columns of the classifier matrix $\mathbf{P}$, provides a set of target prototypes that are maximally and uniformly distant from each other in the binary feature space.

Finding an exact BEF is a hard combinatorial problem. However, we can generate a high-quality approximation using a binary optimization procedure. Given the desired number of classes $C$ and feature dimension $D$, we seek to find the set of vectors $\{\boldsymbol{\rho}_c\}_{c=1}^C$ that minimizes the cost function

$$\mathcal{J}(\{\boldsymbol{\rho}_c\}) = \sum_{i<j} \langle \boldsymbol{\rho}_i, \boldsymbol{\rho}_j \rangle + \alpha \cdot \text{Var}_{i<j}(\langle \boldsymbol{\rho}_i, \boldsymbol{\rho}_j \rangle), \tag{13}$$

where $\text{Var}(\cdot)$ is the variance and $\alpha \geq 0$ balances the two objectives. The first term encourages all pairwise inner products to be negative, while the second pushes them toward a common value.

We optimize this objective using a simple iterative local search algorithm, starting from a random initialization of the $C$ vectors $\{\boldsymbol{\rho}_c\}$ from $\{\pm 1\}^D$. For a fixed number of iterations: *(i)* randomly select a vector $\boldsymbol{\rho}_i$ and a coordinate $k$; *(ii)* compute the change in cost $\Delta\mathcal{J}$ that would result from flipping the sign of the $k$-th element of $\boldsymbol{\rho}_i$; and *(iii)* if $\Delta\mathcal{J} < 0$, accept the flip. This greedy coordinate-wise descent procedure rapidly converges to a local minimum of the cost function. The resulting set of vectors $\{\boldsymbol{\rho}_c\}$ can then be used to construct the fixed classifier matrix $\mathbf{P} = [\boldsymbol{\rho}_1, \ldots, \boldsymbol{\rho}_C]$.

## D  ABLATION STUDIES

### D.1  THE ROLE OF THE GATING THRESHOLD $\nu$ ON MLPs

In this section, we present an ablation study on the gating threshold $\nu$ for binary MLPs, as shown in Figure 4. Empirically, across layer sizes, optimal and non-trivial values of $\nu$ emerge, with the effect becoming more pronounced as NN depth increases, consistent with the RNN ablation in Section 4.4.

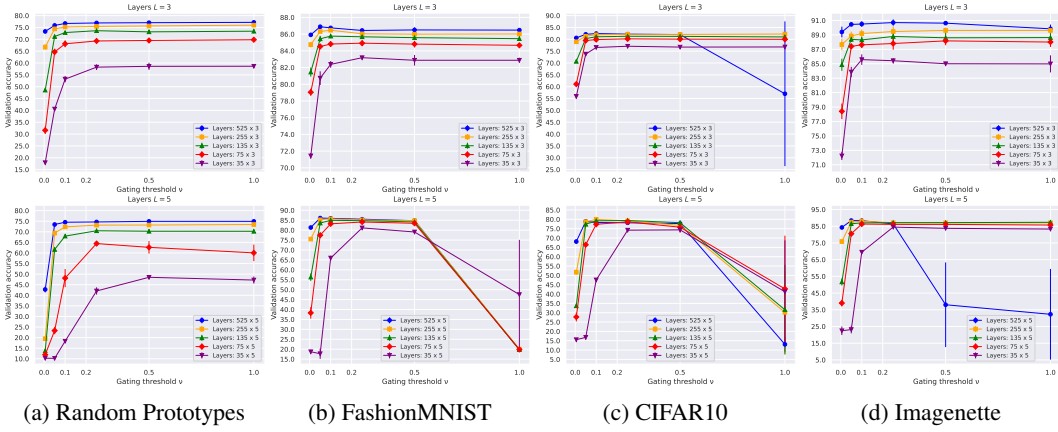

| (a) Random Prototypes | (b) FashionMNIST | (c) CIFAR10 | (d) Imagenette |

Figure 4: Validation accuracy as a function of the gating threshold $\nu$ on Random Prototypes, Fashion-MNIST, CIFAR10, and Imagenette for binary MLPs with $L = 3$ and $L = 5$ hidden layers.

## D.2 THE ROBUSTNESS HYPERPARAMETER $r$

In this section, we perform an ablation study on the robustness parameter $r$ for binary MLPs, as shown in Figure 5. Empirically, setting the margin $r$ in the range 0.5–0.75 consistently yields the best generalization accuracy across the considered tasks and architectures.

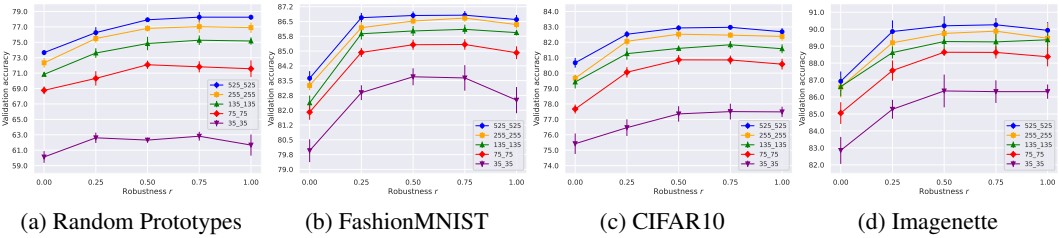

|              (a) Random Prototypes  |  (b) FashionMNIST  |  (c) CIFAR10  |  (d) Imagenette  |

Figure 5: Validation accuracy as a function of the robustness $r$ on Random Prototypes, FashionMNIST, CIFAR10, and Imagenette for binary MLPs with $L = 2$ hidden layers.

## D.3 THE REINFORCEMENT PROBABILITY HYPERPARAMETER $p_r$

In this section, we perform an ablation study on the reinforcement probability $p_r$ for binary MLPs, as shown in Figure 6. Empirically, setting the probability $p_r$ in the range 0.5–0.75 consistently yields the highest generalization accuracy across the evaluated tasks and architectures, although its overall impact remains moderate. Nevertheless, compared to disabling reinforcement entirely, enabling it provides a measurable improvement, with the effect being more marked in lower-capacity models.

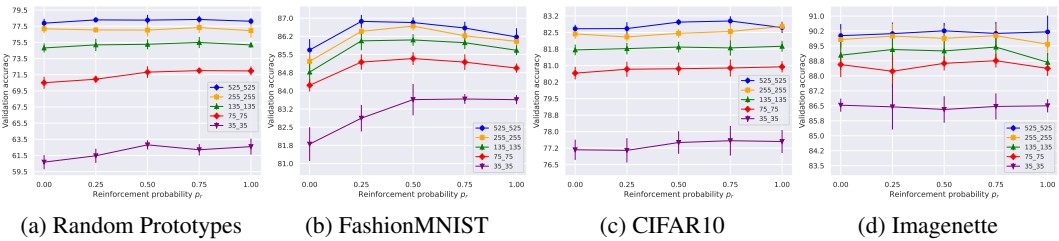

|              (a) Random Prototypes  |  (b) FashionMNIST  |  (c) CIFAR10  |  (d) Imagenette  |

Figure 6: Validation accuracy as a function of the reinforcement probability $p_r$ on Random Prototypes, FashionMNIST, CIFAR10, and Imagenette for binary MLPs with $L = 2$ hidden layers.

## D.4 ANALYSIS OF BINARIZATION FUNCTION

To examine the sensitivity of BEP to the input binarization method, we compare simple median thresholding against thermometer encoding on five UCR datasets using binary RNNs. As shown in Table 3, thermometer encoding significantly outperforms median thresholding on time-series data, as it preserves coarse magnitude information that is essential for these tasks. In contrast, for image data, median thresholding is sufficient to achieve SotA performance with BEP.

Table 3: Validation accuracy on five UCR datasets using different thermometer-encoding bits for input binarization. Notably, using a single bit corresponds to the median-thresholding method.

| Dataset | Thermometer-encoding bits | | | | |
| --- | --- | --- | --- | --- | --- |
| | 1 | 10 | 20 | 50 | 100 |
| *ArticularyWordRec.* | 48.98 ± 3.93 | 81.28 ± 2.99 | 79.37 ± 2.70 | 81.16 ± 2.17 | 81.45 ± 1.85 |
| *DistalPOAgeGroup* | 77.18 ± 2.74 | 78.92 ± 3.38 | 78.11 ± 2.03 | 79.78 ± 2.65 | 79.78 ± 3.74 |
| *ItalyPowerDemand* | 73.54 ± 2.44 | 94.65 ± 1.22 | 94.59 ± 1.13 | 94.34 ± 1.03 | 93.92 ± 1.31 |
| *PenDigits* | 61.60 ± 2.05 | 96.66 ± 0.22 | 96.79 ± 0.35 | 96.88 ± 0.26 | 97.13 ± 0.21 |
| *ProximalPOAgeGroup* | 77.52 ± 3.43 | 82.31 ± 3.28 | 82.37 ± 3.36 | 82.70 ± 2.41 | 82.53 ± 2.85 |

### D.5 ANALYSIS OF MASK DENSITY

To substantiate the claim that the sparsity mask fulfills a role analogous to the learning rate, we evaluate the impact of the group size $\gamma_{0,l}$ (which determines mask density) on training dynamics for binary MLPs. Figure 7 shows that smaller group sizes (sparser updates) lead to slower convergence, whereas larger group sizes (denser updates) accelerate initial learning but may introduce instability. This behavior mirrors the effect of varying the learning rate in gradient-based optimization.

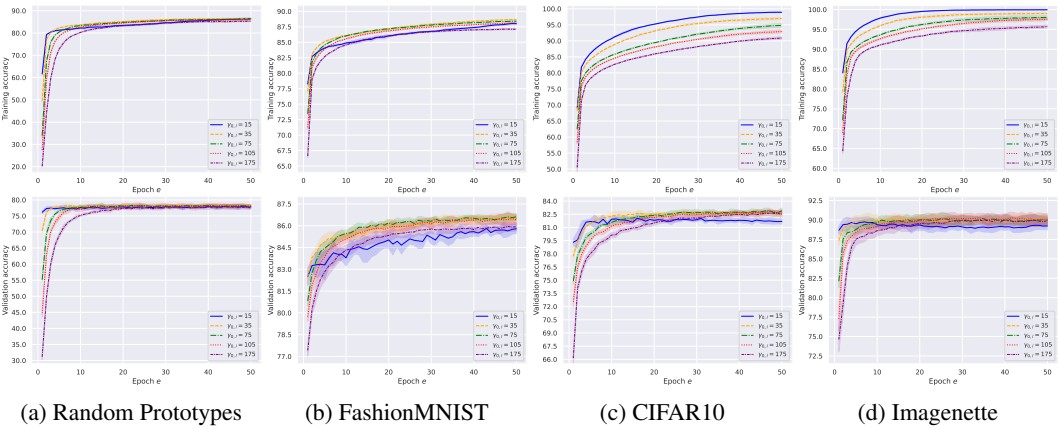

    (a) Random Prototypes      (b) FashionMNIST      (c) CIFAR10      (d) Imagenette

Figure 7: Training and validation accuracy curves over epochs for different values of the group size $\gamma_{0,l}$ on Random Prototypes, FashionMNIST, CIFAR10, and Imagenette.

### D.6 SCALING TO DEEPER NETWORKS

To assess whether BEP can propagate informative error signals through deep computational chains, we evaluate binary RNNs on the S-MNIST dataset while progressively increasing the backward horizon length (i.e., the number of backpropagated time steps). This setup effectively simulates deeper networks by extending the length of the backward computational graph while keeping the temporal window fixed. As shown in Figure 8, performance steadily improves as the backward horizon increases, indicating that BEP successfully propagates useful binary error information over many steps without suffering from excessive signal decay.

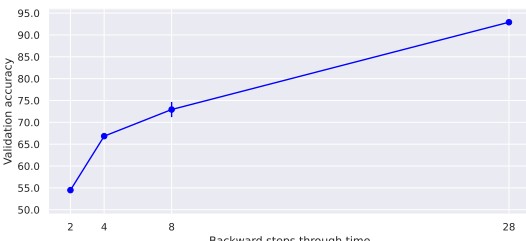

Figure 8: Validation accuracy of a binary RNN trained with BEP on the S-MNIST dataset for different values of the backward horizon length (depth of backpropagation).

