# OpenReview forum: "BEP: A Binary Error Propagation Algorithm for Binary Neural Networks Training"
_ICLR.cc/2026/Conference — ICLR 2026 Poster_

### Official Review · Reviewer_uRee · 2025-10-29

**Soundness:** 3
**Presentation:** 2
**Contribution:** 2
**Rating:** 4
**Confidence:** 2

**Summary:**

This paper presents Binary Error Propagation (BEP), a new training algorithm for binary neural networks (BNNs) where both weights and activations are binary. The key innovation is a fully binary, bitwise analog of the backpropagation chain rule, allowing end-to-end propagation of binary error signals through both feedforward and recurrent architectures. The method is empirically measured against prior binary optimization methods and QAT on several tasks and model architectures, showing test accuracy improvements.

**Strengths:**

1. The method is reasonable. And the formalization of a binary version of global credit assignment for BNNs is clear.
2. The empirical results are broad, showing the advantage of BEP.

**Weaknesses:**

1. The experiments are not sufficient. Although the authors conduct experiments on both MLP and RNN models to demonstrate the effectiveness of their method, the experimental setup appears somewhat toy. Could the authors perform experiments on larger-scale networks (e.g., comparable to ResNet in size) and datasets (e.g., ImageNet-1k)? In addition, since the BNN field has been studied for a long time, could the authors compare BEP with other BNN methods (like [1-3]) beyond QAT to further validate its effectiveness?

[1] Bi-Real Net: Enhancing the Performance of 1-bit CNNs With Improved Representational Capability and Advanced Training Algorithm

[2] MeliusNet: Can Binary Neural Networks Achieve MobileNet-level Accuracy?

[3] XNOR-Net++: Improved Binary Neural Networks

2. Have the authors measured the actual training overhead, such as training time and memory usage, to demonstrate that BEP indeed achieves better training efficiency compared to the baseline?

3. The paper format does not appear to follow the ICLR style. The pages seem wider and contain more content, which might be unfair to other submissions.

**Questions:**

1. Similar to Weakness 1: Can BEP be applied to deeper binary CNNs or transformer-style architectures without encountering collapse or stagnation? Have the authors attempted to apply BEP to modern vision or language models beyond the provided MLP/RNN settings?

2. Batch normalization seems to contribute significantly to performance improvement. Have the authors considered adopting a binary analog or an alternative form of normalization within BEP?

---

> ### Author Response · Authors · 2025-11-24
> **Detailed Comments (1)**
>
> We thank the reviewer for their insightful comments. We have revised the manuscript to address all raised concerns. Below is a point-by-point response to the weaknesses (W) and questions (Q).
>
> - **W1,Q1**: We agree that extending BEP to larger-scale architectures such as ResNet- or transformer-sized models, as well as to datasets like ImageNet-1k, is an important direction for future work. Running BEP on these pipelines would require a full CNN/transformer instantiation of our method, including binary convolutions, transposed convolutions for backward propagation, filter-level masking, and appropriate architectural adaptations. As discussed in the new Discussion and Limitations section (Section 4.5), we view this as a natural but technically non-trivial extension and a core direction of our ongoing research. In the present study our primary objective is to introduce, characterize, and experimentally validate a new fully binary training rule that replaces the floating-point-based forward and backward passes with purely binary computations. To this end, we evaluated BEP on FashionMNIST, CIFAR-10 and Imagenette using multi-layer MLPs, and on S-MNIST and 30 UCR time-series datasets using RNNs, consistently outperforming both QAT-based training and the prior fully binary local learning rule by Colombo et al. (2025), which, to the best of our knowledge, is the only other method performing binary training. Moreover, we added five ablation studies in the Appendix to provide a more in-depth analysis of BEP: (i) the robustness parameter $r$, (ii) the reinforcement probability $p_r$, (iii) the binarization function, (iv) the mask density, and (v) the scaling to deep backpropagation chains.
> To further clarify the scope of this work and better position BEP within the broader BNN landscape, we have expanded the Related Work sections to discuss recent BNN methods (including Bi-Real Net, MeliusNet, and XNOR-Net++ [1–3]) and to explain how they relate to our fully binary approach. Specifically, we emphasize that all existing binary CNN methods are trained using QAT, which relies on full-precision SGD/Adam with latent real-valued weights and gradients. Consequently, these approaches cannot benefit from the computational and memory efficiency that our solution provides, particularly in the backward pass, where binary operations replace more costly full-precision computations. Moreover, state-of-the-art BNN architectures typically include components such as batch normalization, full-precision input/output layers, and skip connections, which prevent them from being fully binary even at inference time, making them not directly comparable to our solution. In contrast, BEP maintains binary computation throughout both training and inference, leading to substantial gains in memory and computational efficiency.
> Lastly, we also mentioned in the Conclusion section potential application domains, such as tiny ML, privacy-preserving ML, and neuromorphic hardware, where our fully binary training rule would be particularly suitable.

---

> ### Author Response · Authors · 2025-11-24
> **Detailed Comments (2)**
>
> - **W2**: To address this point, we have added a new comparative table in Section 4.5 (Table 2), which contrasts our method with related work in terms of memory footprint and computational cost. Specifically, the table reports the number of bits required to store weights, activations, desired activations (i.e., error signals), and weight updates, as well as the computational complexity of the forward, backward, and weight-update passes, expressed in terms of the number of operations. Compared to QAT-based methods that rely on floating-point gradients, our approach achieves a $2\times$ reduction in memory usage for hidden weights and a $32\times$ reduction for error signals and weight updates. Moreover, Adam-based QAT requires an additional $64$ bits per parameter to store its first- and second-order moment estimates, further increasing its memory demand.
> Regarding computation, we emphasize that all operations in our training algorithm are binary and can be naturally expressed using Boolean logic primitives such as XNOR, Popcount, and increment/decrement operations. To quantify computational cost at the hardware level, we adopt the equivalent-Boolean-gate metric from Colombo et al. (2025) which compares the intrinsic complexity of different arithmetic operations. This metric provides a more accurate basis for comparison than simply counting high-level operations or FLOPs. While reporting the number of operations alone is possible, it would be misleading, as IEEE-754 single-precision (32-bit) floating-point additions and multiplications [4] used by QAT are estimated to require on the order of $10^4$ equivalent Boolean gates [5], whereas N-bit Popcount and increment/decrement operations on N-bit integers require at most O(10N) Boolean gates [6]. Consequently, a single floating-point addition or multiplication can be up to three orders of magnitude more costly than a binary operation in terms of gate count, and therefore in time, space, and power. Summarizing, from a computational-complexity perspective, we observe roughly a $1000\times$ reduction in the number of Boolean-gate operations required for error-signal backpropagation and weight updates compared with QAT.
> This addition clarifies the specific aspects of BNN training that our method targets and improves, namely, memory demand and computational cost. It also provides a clearer understanding of the potential impact of our approach on overall resource usage.
>
> - **W3**: We thank the reviewer for bringing this to our attention. We discovered that a LaTeX package unintentionally modified the page margins. We apologize for this oversight. In the revised manuscript, we have removed the package and restored the correct ICLR formatting. We would like to emphasize that all content in the paper fits within the original 9 page limits prescribed by the conference. Even without altering the scientific material, the manuscript remains compliant by adjusting figure sizes (Fig.1 and 3) and placing certain equations in-line rather than displayed.
>
> - **Q2**: We agree that Batch Normalization (BN) can improve the performance of BNNs, which is why we reported QAT baselines both with and without BN for reference. In the current work, however, we deliberately do not incorporate BN into BEP, as our goal is to isolate and analyze the proposed fully binary training rule while keeping the entire training and inference pipeline in the binary domain. Standard BN is typically implemented and trained in the real-valued domain, which would partially compromise this objective by introducing floating-point components into the solution.
> Nonetheless, BEP is compatible with normalization mechanisms. Conceptually, it is possible to augment BEP with small real-valued normalization layers on top of the binary pathway, or design discrete analogs (such as integer offsets or layerwise scale factors) that adjust pre-activations while preserving the binary nature of weights and activations. We have clarified this design choice in the revised manuscript and now explicitly list BN-style or offset-based normalization strategies as promising extensions in the new Discussion and Limitations section (Section 4.5).
>
> [4]: 754-2019 - IEEE Standard for Floating-Point Arithmetic, 2019
>
> [5]: Convolutional differentiable logic gate networks, 2024
>
> [6]: Differentiable Weightless Neural Networks, 2024

---

> ### Author Response · Authors · 2025-12-03
> **Summary for ACs**
>
> Here we provide a summary of our responses to the principal points raised by the reviewer for the Area Chairs, while detailed point-by-point replies are included above. The revised manuscript addresses all of the reviewer's suggestions and requests for clarifications and additional analyses as follows:
>
> 1. **We expanded our experimental analysis and strengthened the positioning of BEP within the broader BNN literature to address concerns regarding the scale and scope of our evaluations, as detailed in the new Discussion and Limitations section (Section 4.5)**. In the revised manuscript, we clarified that the goal of this work is to introduce and validate the first fully binary backpropagation rule using MLPs and RNNs. To reinforce the empirical foundation, we added five additional ablation studies in the Appendix covering: *(i)* the robustness parameter $r$, *(ii)* the reinforcement probability $p_r$, *(iii)* the binarization function, *(iv)* the mask density, and *(v)* scaling to deep backpropagation chains. We also expanded the Related Work section to include recent BNN architectures and clarified why these QAT-based models are not directly comparable to BEP: they rely on real-valued gradients, batch normalization, and other full-precision components, whereas BEP maintains binary computation throughout training and inference. Finally, we highlighted application domains such as TinyML, privacy-preserving ML, and neuromorphic hardware, where fully binary training is especially advantageous.
>
> 2. **We added a new comparative table (Table 2 in Section 4.5) reporting memory usage and computational cost** across all relevant components (weights, activations, desired activations, and weight updates) and contrasting them with QAT-based methods. Our approach achieves a $2\times$ reduction in memory usage for hidden weights and a $32\times$ reduction for error signals and weight updates. We further adopted the equivalent-Boolean-gate metric from Colombo et al. (2025), which provides a hardware-level comparison: whereas IEEE-754 floating-point operations used in QAT require on the order of $10^4$ Boolean gates, binary XNOR, Popcount, and increment/decrement operations scale as $\mathcal{O}(10N)$. This demonstrates an estimated $1000\times$ reduction in gate-level complexity for BEP’s backward and update computations.
>
> 3. **We added a comprehensive Discussion and Limitations section (Section 4.5)** that clearly articulates BEP’s current constraints and outlines future directions. This section now discusses limitations in architectural scope (extending BEP to fully binary CNNs or transformers), task scope (binary segmentation, multi-label prediction, and other binary-output tasks), and scale (application to ImageNet-level datasets or very deep models). Specifically, BEP extends naturally to convolutional layers by treating filters as binary perceptrons applied to local input patches, yielding binary and transposed binary convolutions. Filter updates aggregate Hebbian contributions across all receptive fields, and sparsity can be enforced via filter- or channel-level masking. Although technically challenging, these extensions are conceptually compatible with BEP and are part of our ongoing research efforts. We also clarified that BEP scales effectively to longer computational chains, as demonstrated by the new depth-scaling experiment in Appendix D.6. Finally, we added clarification regarding normalization mechanisms. Although we intentionally avoided batch normalization to maintain a fully binary training pipeline, we now explain that BEP is compatible with lightweight real-valued normalization layers or discrete normalization analogues (e.g., integer offsets, layerwise scale factors). These strategies are now explicitly listed as promising future extensions.

---

### Official Review · Reviewer_hAH5 · 2025-10-30

**Soundness:** 3
**Presentation:** 3
**Contribution:** 3
**Rating:** 6
**Confidence:** 4

**Summary:**

The paper introduces Binary Error Propagation (BEP), a new training algorithm for binary neural networks that avoids surrogate gradients. Instead of real-valued backpropagation, BEP defines a discrete backward rule that propagates binary "desired activations" using only bitwise operations. The method updates integer hidden weights through sparse binary masks that act as an adaptive learning mechanism. BEP is evaluated on MLPs and RNNs across several datasets, showing higher accuracy than quantization-aware training (QAT) and the method proposed in Colombo et al. (2025). The authors claim that this is the first fully binary error Backpropagation (BP) algorithm capable of effectively training BNNs without relying on floating-point gradients.

**Strengths:**

- The paper is clearly written and well organized, with notation that is consistent and easy to follow.
- The presentation is concise and direct, and I found the mathematical derivations correct and well-grounded.
- The method is novel and well motivated, and the experimental results are consistent with most of the claims.

**Weaknesses:**

- In Section 3.1, the authors obtain binary input representations using a fixed binarization method (median or thermometer encoding) but do not analyze how different binarization functions affect BEP's performance.

- The margin parameter $r$ in Eq. (1) determines when binary updates are triggered, but its value and sensitivity are not analyzed. Since this parameter effectively affects the learning dynamics, the authors should provide an ablation to show how it influences performance.

- The paper asserts that the sparsity mask $M_l$ fulfills the role of the learning rate in classical BP, yet this is only mentioned in the text. Providing empirical evidence would strengthen this claim.

- Experiments are limited to two- and three-layer MLPs. It would be helpful to discuss whether BEP scales to deeper binary networks, as the discrete backward propagation might face stability issues analogous to vanishing gradients.

- The paper should include a limitations section. While the authors briefly mention possible future directions, they should clearly articulate the current constraints of BEP.

Minor comments:
- The related work section would benefit from including recent works such as "BiPer: Binary Neural Networks using a Periodic Function (CVPR 2024)"
- The small QAT (w/o batchnorm) plots embedded in each subfigure of Figure 2 are difficult to interpret. I recommend improving their presentation or visibility for better clarity.
- In sections 4.3 and 4.4, the term "window length" is not clearly defined.

**Questions:**

- Have you tested other binarization functions, and can you comment on how sensitive BEP is to this choice?
- The margin parameter $r$ in Eq. (1) controls when updates are triggered. Have you analyzed how different $r$ values affect training stability or accuracy?
- You state that the sparse mask $M_l$ plays a role similar to a learning rate. Could you provide additional analysis to support this interpretation?
- Have you tested BEP on deeper binary MLPs? If not, could you comment on potential challenges in extending BEP to deeper networks?
- Given that most prior work on BNNs targets convolutional architectures, can you elaborate on how BEP might extend to CNNs?
- Did you test BEP on tasks other than classification? Might it work?

---

> ### Author Response · Authors · 2025-11-24
> **Detailed Comments (1)**
>
> We thank the reviewer for their insightful comments. We have revised the manuscript to address all raised concerns. Below is a point-by-point response to the weaknesses (W) and questions (Q).
>
> - **W1,Q1**: In our framework, the binarization step is orthogonal to BEP itself. The algorithm only requires binary inputs $a_0 = \text{bin}(x) \in {\pm 1}^{K_0}$, and different binarization functions primarily affect how much task-relevant information is preserved in the input rather than the behavior of the training rule. For image data, we use simple per-pixel median binarization, which already yields the generalization performance reported in the paper. For time-series and tabular data, we rely on thermometer encoding because it preserves coarse magnitude information, which is crucial for the UCR and other sequence-classification tasks.
> In the revised manuscript, we include an ablation comparing median threshold-based binarization and thermometer encoding on five representative UCR datasets in the Appendix (Appendix D.4). This experiment shows that BEP is robust to reasonable binarization schemes but clearly benefits from thermometer encoding in time-series settings where magnitude carries critical information.
>
> - **W2,Q2**: In response to this suggestion, we expanded our analysis by conducting additional ablation experiments on the sensitivity of the key margin hyperparameter $r$. These results are now included in the Appendix (Appendix D.2). The robustness margin $r$ enforces a larger separation between the predicted class and the remaining classes at the classifier level. This induces more robust binary error signals and improves generalization by encouraging the model to learn representations with wider decision boundaries. Empirically, we found that setting the margin r in the range 0.5--0.75 consistently yields the best generalization accuracy across the considered tasks and architectures.
>
> - **W3,Q3**: To substantiate the claim, we have added an empirical analysis in the Appendix (Appendix D.5), where we vary the sparsity mask across a range of values and report the corresponding training and validation curves. These experiments show that the sparsity mask directly controls the speed of convergence. Denser masks produce faster error reduction, whereas sparser masks slow down learning. This behavior closely mirrors the effect of the learning rate in classical backpropagation.
> We also clarify that, although the sparsity mask is not a learning rate in the strict mathematical sense, it serves an analogous functional role in BEP. In backpropagation, the learning rate scales the magnitude of continuous parameter updates; in BEP, each update has fixed magnitude $\pm$ 1, and the mask instead modulates where updates occur, effectively regulating the spatial locality and overall intensity of learning. In this sense, the sparsity mask governs the convergence dynamics of the algorithm in a manner similar to how the learning rate regulates convergence in gradient-based methods.
>
> - We have addressed all three minor comments in the revised manuscript. Specifically, we have included the suggested reference and enhanced the visibility of the plots in Figure 2. Moreover, we have clarified the definition of “window length” in Sections 4.3 and 4.4, specifying that it refers to the number of timesteps included in the temporal window used for each dataset in the RNN experiments.

---

> ### Author Response · Authors · 2025-12-03
> **Detailed Comments (2)**
>
> - **W4,Q4**: To clarify whether BEP scales to deeper binary networks, we have expanded the manuscript accordingly. First, we added a new experiment demonstrating that BEP can train deep backpropagation chains. In MLP experiments, empirical evidence shows that increasing depth beyond two or three hidden layers yields saturated accuracy for many real-world tasks (both with BEP and with full-precision optimizers such as SGD and Adam [1-3]). To further investigate BEP’s ability to propagate useful information across many layers, we therefore conducted an additional analysis in the Appendix (Appendix D.6) on the S-MNIST dataset using an RNN. By progressively increasing the number of backpropagated time steps (while keeping the temporal window fixed), we effectively deepen the computational graph. The results show that performance steadily improves as the backward horizon grows, demonstrating that BEP is capable of transmitting useful binary error information across long computational chains.
> This behavior is supported by the design of BEP’s update rule. Beyond the boundedness of the $\pm 1$ updates, BEP employs a gating mechanism that selectively controls which neurons participate in the backward pass. This gate allows only neurons whose pre-activations lie near the decision boundary (i.e., $|z_{l+1}^\mu| \le \nu K_l$) to propagate their binary error signals, while saturated units are blocked. This prevents ineffective or noisy updates from propagating through the network, thereby avoiding stability issues analogous to gradient vanishing. The ablation study in Section 4.4 shows that this gating mechanism enables stable and effective learning over long sequences with RNNs. Very small or very large gating thresholds degrade accuracy, while intermediate values yield robust behavior. This demonstrates that BEP’s discrete gating mechanism prevents instability in long-range temporal backward propagation.
> Nonetheless, we agree that a systematic investigation of depth scaling is an important question. As part of our ongoing work, we are exploring architectural strategies such as incorporating skip connections to reinforce the backward signal. Since BEP’s forward and backward passes operate on binary signals, a skip connection simply introduces an additional binary error pathway that is merged via elementwise addition (followed by sign projection) at the receiving layer. These points are now explicitly discussed in the new Discussion and Limitations section (Section 4.5).
>
> - **W5**: In the revised manuscript, we have added a dedicated Discussion and Limitations section (Section 4.5) that clearly articulates the current constraints of BEP, highlights the main challenges for extending the method, and outlines possible directions for future work. This section integrates the various points raised across the review and clarifies the present scope of BEP:
> 1. Architecture: Our study focuses on binary MLP and RNN architectures. Extending BEP to convolutional or transformer-style models requires a full binary redesign of these architectures, including binary convolutions, transposed convolutions for the backward pass, filter-level masking, and other architectural adaptations to support weight sharing, spatial structure, and multi-head mechanisms. This extension is technically non-trivial and constitutes a core direction of our ongoing research.
> 2. Task and output: The current experiments focus on classification tasks. While BEP naturally supports arbitrary binary output vectors and can, in principle, be applied to multi-label prediction, binary segmentation, or more general binary-vector regression, these tasks require additional design choices for the output encoding. We now explicitly list these as promising extensions.
> 3. Dataset scale and model depth: Our empirical evaluation spans FashionMNIST, CIFAR-10, Imagenette, S-MNIST, and 30 UCR time-series datasets using moderate-depth MLPs and RNNs. Applying BEP to large-scale datasets and to very deep models (e.g., ImageNet-level CNNs) would require substantial design adaptations specific to convolutional or transformer architectures. We emphasize this as an important extension of the present work. Moreover, the introduction of binary-compatible normalization mechanisms may become necessary to enable stable BEP training in such large-scale settings.
>
> [1] Deep learning, 2015
>
> [2] Do deep nets really need to be deep?, 2014
>
> [3] Training multi-layer binary neural networks with random local binary error signals, 2025

---

> ### Author Response · Authors · 2025-12-03
> **Detailed Comments (3)**
>
> - **Q5**: We agree that extending BEP to convolutional architectures is an important direction. However, we emphasize that existing CNN-based BNNs are all trained using QAT, which relies on full-precision SGD/Adam with latent real-valued weights and gradients. Consequently, these methods cannot benefit from the computational and memory efficiency that our solution provides, particularly in the backward pass, where binary operations replace more costly full-precision computations. Moreover, because such architectures typically include components like batch normalization, full-precision input/output layers, and skip connections, they are not fully binary even at inference time.
> Conceptually, BEP is defined for generic linear layers, and a convolutional filter can therefore be viewed as a binary perceptron applied to local input patches. Under this interpretation, the forward and backward passes become binary convolutions and transposed binary convolutions, while the update rule in Eq. (8) corresponds to updating filter stabilities by aggregating the Hebbian outer-product contributions across all spatial locations where the filter is applied. Weight sharing is naturally captured because each filter receives stability updates from all of its receptive fields, and the masking mechanism in Eq. (9) can be applied at the filter or channel level to keep updates sparse. As discussed in the new Discussion and Limitations section (Section 4.5), this constitutes a core direction of our ongoing research.
>
> - **Q6**: In the current work we evaluated BEP only on classification tasks (both image and time-series). Conceptually, however, BEP requires only binary target activations at the output layer and therefore can, in principle, extend to any task, given that targets are represented as binary vectors. This includes binary-vector regression tasks such as binary segmentation, multi-label prediction, and attribute estimation. For example, binary segmentation can be formulated as predicting a high-dimensional binary mask that is directly supplied as the desired activation pattern at the output layer. In this formulation, BEP operates exactly as in classification but with a larger binary output space, effectively bypassing the need for a separate decoder. Similarly, multi-label classification tasks can encode each attribute as a bit in a binary target vector, allowing BEP to train in the same way as in our current setting. We now clarify in the revised manuscript that extending BEP to binary segmentation, multi-label prediction, and more general binary-vector regression is a natural and promising direction for future work, given the method’s compatibility with arbitrary binary output structures.

---

> ### Author Response · Authors · 2025-12-03
> **Summary for ACs**
>
> Here we provide a summary of our responses to the principal points raised by the reviewer for the Area Chairs, while detailed point-by-point replies are included above. The revised manuscript addresses all of the reviewer's suggestions and requests for clarifications and additional analyses as follows:
>
> 1. **We added an ablation study on binarization functions (Appendix D.4)** comparing median thresholding and thermometer encoding on five representative UCR datasets, showing that BEP is robust to reasonable binarization choices, while thermometer encoding is beneficial for time-series tasks where magnitude information is critical. We also clarified in the main text that binarization is orthogonal to the BEP algorithm and primarily influences the informativeness of the binary input representation.
>
> 2. **We expanded our hyperparameter analysis with new ablations for the robustness margin parameter $r$ (Appendix D.2)**, which controls when binary updates are triggered. These experiments show that values of $r$ within $0.5$–$0.75$ consistently yield the best generalization by encouraging larger decision margins and more reliable binary error signals.
>
> 3. **We provided empirical evidence demonstrating the functional role of the sparsity mask (Appendix D.5)**. By varying the density of the mask, we show how it controls the speed of convergence: denser masks accelerate learning, while sparser masks slow it down. This mirrors the effect of the learning rate in classical backpropagation, with mask density regulating where BEP updates occur rather than their magnitude.
>
> 4. **We introduced a new experiment evaluating BEP’s ability to propagate error signals across deeper computational chains (Appendix D.6)**. Using an RNN on the S-MNIST dataset, we progressively increased the backward horizon (while keeping the window length fixed), effectively deepening the model, and observed monotonic improvements in performance. This demonstrates that BEP can transmit informative binary error signals over long ranges. We also clarified how BEP’s gating mechanism prevents discrete analogues of vanishing gradients by allowing only neurons near the decision boundary to propagate error signals.
>
> 5. **We added a dedicated Discussion and Limitations section (Section 4.5)** to clearly articulate BEP’s current constraints and outline future directions. This section now discusses limitations in architectural scope (extending BEP to fully binary CNNs or transformers), task scope (binary segmentation, multi-label prediction, and other binary-output tasks), and scale (application to ImageNet-level datasets or very deep models). Specifically, BEP extends naturally to convolutional layers by treating filters as binary perceptrons applied to local input patches, yielding binary and transposed binary convolutions. Filter updates aggregate Hebbian contributions across all receptive fields, and sparsity can be enforced via filter- or channel-level masking. Although technically challenging, these extensions are conceptually compatible with BEP and are part of our ongoing research efforts. For non-classification tasks, we added an explicit discussion explaining that BEP requires only binary desired activations at the output layer and therefore extends naturally to multi-label prediction, binary segmentation, and more general binary-vector regression problems.

---

### Official Review · Reviewer_N7bj · 2025-11-01

**Soundness:** 3
**Presentation:** 2
**Contribution:** 3
**Rating:** 6
**Confidence:** 4

**Summary:**

The work proposes a novel alternative to the backpropagation algorithm by reformulating it in the binary domain to reduce computational cost. Unlike Quantization-Aware Training, BEP performs both the forward and backward passes entirely in the binary domain, relying solely on bitwise operations. The paper provides theoretical foundations and experimental results that validate this reformulation of backpropagation.

**Strengths:**

Novelty: the paper introduces an alternative approach to regular backpropagation by reformulating the algorithm to work for binary weights.


Different from other approaches that rely on real value parameters, like a straight-through estimator, this work focuses on the idea of computing forward and backward on the binary domain.


The work displays consistent improvements over the classification task on two datasets.

**Weaknesses:**

1. Lack of organization, it is difficult to read smoothly, as figures, such as Figure 2, are on page 7, and mentioned in page 8, same as Table 1, where it is mentioned in page 9, and it is on page 8.


2. The authors claim computational efficiency and memory reduction, but ablation studies over flops, training, and inference time, as well as details over the computational equipment, are not stated.


3. The authors propose several hyperparameters, even though in section 4.4 is an analysis. Further ablation studies should be conducted on the sensitivity of parameters like r and pr.


4. Lack of training details hinders the reproducibility of the work, as it is not stated the epochs or the number of iterations used in the experiments.


5. The update rule should be further clarified. Even though there are references to previous works, it should be explicitly stated where they come from Eq. (8)


6. The binary mask from Equation 9 is not theoretically explained, and its purpose is not clear.

**Questions:**

How does BEP scale computationally compared to QAT during training?


1. How sensitive is BEP to initialization and hyperparameter selection across datasets?


2. Does the algorithm have similar results across different tasks to classification, for instance, binary segmentation?


3. How does BEP handle vanishing or exploiting gradients in comparison to the standard backpropagation algorithm?


4. How does the algorithm work for larger spatial dimensions and complex feature datasets, such as the celebA dataset with the multilabel classification task?

---

> ### Author Response · Authors · 2025-11-24
> **Detailed Comments (1)**
>
> We thank the reviewer for their insightful comments. We have revised the manuscript to address all raised concerns. Below is a point-by-point response to the weaknesses (W) and questions (Q).
>
> - **W1**: To address the layout and organization issues, we have carefully revised the manuscript to ensure that all figures and tables (e.g., Figure 2 and Table 1) appear close to the corresponding text where they are first referenced.
>
> - **W2,Q0**: To address this point, we have added a new comparative table in Section 4.5 (Table 2), which contrasts our method with related work in terms of memory footprint and computational cost. Specifically, the table reports the number of bits required to store weights, activations, desired activations (i.e., error signals), and weight updates, as well as the computational complexity of the forward, backward, and weight-update passes, expressed in terms of the number of operations. Compared to QAT-based methods that rely on floating-point gradients, our approach achieves a $2\times$ reduction in memory usage for hidden weights and a $32\times$ reduction for error signals and weight updates. Moreover, Adam-based QAT requires an additional $64$ bits per parameter to store its first- and second-order moment estimates, further increasing its memory demand.
> Regarding computation, we emphasize that all operations in our training algorithm are binary and can be naturally expressed using Boolean logic primitives such as XNOR, Popcount, and increment/decrement operations. To quantify computational cost at the hardware level, we adopt the equivalent-Boolean-gate metric from Colombo et al. (2025) which compares the intrinsic complexity of different arithmetic operations. This metric provides a more accurate basis for comparison than simply counting high-level operations or FLOPs. While reporting the number of operations alone is possible, it would be misleading, as IEEE-754 single-precision (32-bit) floating-point additions and multiplications [1] used by QAT are estimated to require on the order of $10^4$ equivalent Boolean gates [2], whereas N-bit Popcount and increment/decrement operations on N-bit integers require at most $\mathcal{O}(10N)$ Boolean gates [3]. Consequently, a single floating-point addition or multiplication can be up to three orders of magnitude more costly than a binary operation in terms of gate count, and therefore in time, space, and power. Summarizing, from a computational-complexity perspective, we observe roughly a $1000\times$ reduction in the number of Boolean-gate operations required for error-signal backpropagation and weight updates compared with QAT.
> This addition clarifies the specific aspects of BNN training that our method targets and improves, namely, memory demand and computational cost. It also provides a clearer understanding of the potential impact of our approach on overall resource usage.
> All experiments were conducted on an Ubuntu 20.04 LTS workstation with 2 Intel Xeon Gold 5318S CPUs, 384 GB of RAM, and an Nvidia A40 GPU.
>
> - **W3,Q1**: In response to this suggestion, we expanded our analysis by conducting additional ablation experiments on the sensitivity of the key hyperparameters $r$ and $p_r$. These results are now included in the Appendix (Appendix D.2-D.3), where we report the impact of varying each parameter independently.
> The robustness margin $r$ enforces a larger separation between the predicted class and the remaining classes at the classifier level. This induces more robust binary error signals and improves generalization by encouraging the model to learn representations with wider decision boundaries. Conversely, the reinforcement probability $p_r$ strengthens the existing memory trajectories of each integer weight, with its value adaptively rescaled at each epoch based on the training errors and normalized by the layer width. This increases reinforcement when the model is uncertain and balances its effect across layers of different sizes. Empirically, we found that setting the margin $r$ in the range $0.5$--$0.75$ consistently yields the best generalization accuracy across the considered tasks and architectures. Conversely, the reinforcement probability $p_r$ has a moderate impact on generalization. Nevertheless, compared to disabling reinforcement entirely, enabling it provides a measurable improvement, with the effect being more marked in lower-capacity models.
> Lastly, to address sensitivity to initialization, we repeated all experiments using multiple random seeds. In addition, for the UCR experiments, we performed 3-fold validation to obtain more statistically robust results. For each configuration, we report the mean and standard deviation across runs, providing a reliable estimate of the stability of the method under different initial conditions.
>
> [1]: 754-2019 - IEEE Standard for Floating-Point Arithmetic, 2019
>
> [2]: Convolutional differentiable logic gate networks, 2024
>
> [3]: Differentiable Weightless Neural Networks, 2024

---

> ### Author Response · Authors · 2025-11-24
> **Detailed Comments (2)**
>
> - **W4**: In response to this observation, we have updated Section 4 to include the full training details for all experiments. This ensures that all reported results can be precisely reproduced. Additionally, we emphasize that we will release the complete source code and configuration files upon acceptance of the manuscript to ensure full reproducibility.
>
> - **W5**: We have clarified the origin and interpretation of the update rule in Eq. (8). Specifically, Eq. (8) is a direct matrix generalization of the classical perceptron update [4] and, more precisely, of the Clipped Perceptron (CP) and CP+Reinforcement (R) rules [5,6], which correspond to supervised Hebbian-like outer-product updates. For a single output neuron, $\Delta H_l^\mu = a_l^{\*\mu}(a_{l-1}^\mu)^\top$ simply adds (or subtracts) the input vector $a_{l-1}^\mu$ to the synaptic stability variables $H_l$ whenever the desired output $a_l^{\*\mu}$ is +1  (or -1), exactly as in the original CP rule. The multi-layer version presented in Eq. (8) applies the same outer-product mechanism row-wise (one binary perceptron per hidden unit), where the integer variables $H_l$ play the role of metaplastic synaptic variables.
> In the revised manuscript, we have expanded the explanation surrounding Eq. (8) to make this connection explicit and to ensure that the origin and interpretation of the update rule are clearly stated.
>
> - **W6**: We have clarified the purpose and theoretical motivation of the binary mask introduced in Eq. (9). The mask $M_l^\mu$ is used to sparsify updates at the neuron level and to focus learning on the least stable units. For each group of neurons in layer $l$, Eq. (9) selects the neuron with the smallest signed stability $a_l^{\*\mu} H_{l,j}$ (i.e., the neuron closest to misclassification) and sets its corresponding mask entry to 1. As a result, $\Delta H_l^\mu = M_l^\mu \odot \bigl(a_l^{\*\mu} (a_{l-1}^\mu)^\top\bigr)$ updates only that neuron, while the others receive no update. This mechanism serves as a discrete, data-dependent form of learning-rate control. It bounds the number of synapses updated per pattern and prevents over-reinforcing neurons that already classify the pattern with high confidence. This behavior is consistent with CP+R and SBPI-style supervised Hebbian learning rules, where updates are intentionally directed toward the most unstable or error-prone units.
> In the revised manuscript, we make this explanation explicit immediately after Eq. (9). In addition, we have included empirical curves in the Appendix (Appendix D.5) showing how varying the mask density (via the group size) trades off convergence speed and generalization accuracy, further confirming its role as an effective sparsifying and learning-rate-like mechanism.
>
> - **Q2**: BEP requires only binary target activations at the output layer, and therefore can, in principle, extend to any task, given that targets are represented as binary vectors. Binary segmentation, for example, can be formulated as predicting a high-dimensional binary mask, which can be directly supplied as the desired activation pattern at the output layer. In this formulation, BEP would operate exactly as in classification but with a larger binary output space, effectively bypassing the need for a separate decoder. We now clarify in the revised manuscript that binary segmentation, and, more broadly, binary-vector regression, is a promising application direction for future work, given the compatibility of BEP with arbitrary binary output structures.
>
> [4] The perceptron: A probabilistic model for information storage and organization in the brain, 1958
>
> [5] Efficient supervised learning in networks with binary synapses, 2007
>
> [6] Generalization Learning in a Perceptron with Binary Synapses, 2009

---

> ### Author Response · Authors · 2025-11-24
> **Detailed Comments (3)**
>
> - **Q3**: In BEP, the issues typically associated with vanishing or exploding gradients do not arise in the same form as in standard backpropagation, because BEP does not propagate real-valued gradient magnitudes. Instead, the backward pass consists of binary desired activations, and the updates to the synaptic stability variables $H_l$ occur in fixed $\pm 1$ steps within a bounded integer range. As a result, there is no mechanism for unbounded growth of update magnitudes (“gradient explosion”), and the signal does not diminish continuously as in floating-point backpropagation (“gradient vanishing”). Beyond the boundedness of the update rule, BEP employs a gating mechanism that selectively controls which neurons participate in the backward pass. This gate allows only neurons whose pre-activations lie near the decision boundary (i.e., $|z_{l+1}^\mu| \le \nu K_l$) to propagate their binary error signals, while saturated units are blocked. This prevents ineffective or noisy updates from propagating through the network.
> The ablation study in Section 4.4 shows that this gating mechanism enables stable and effective learning over long sequences with RNNs. Very small or very large gating thresholds degrade accuracy, while intermediate values yield robust behavior. This demonstrates that BEP’s discrete gating mechanism leads to stable depth-wise signal propagation without suffering from attenuation and instability.
>
> - **Q4**: As now discussed in Section 4.5, BEP is agnostic to the spatial structure of the input and only requires binary target activations at the output layer. In principle, this makes it applicable to high-resolution and multi-label tasks such as CelebA. For multi-label classification, each attribute can be represented as a bit in a binary target vector, allowing BEP to operate exactly as in our current setting by using this binary vector as the desired activation pattern at the output layer. We now clarify in the revised manuscript that extending BEP to CelebA-style multi-label scenarios is a natural and promising direction for future work, given the method’s compatibility with arbitrary binary output structures.

---

> ### Author Response · Authors · 2025-12-03
> **Summary for ACs**
>
> Here we provide a summary of our responses to the principal points raised by the reviewer for the Area Chairs, while detailed point-by-point replies are included above. The revised manuscript addresses all of the reviewer's suggestions and requests for clarifications and additional analyses as follows:
>
> 1. **We added a new comparative table (Table 2 in Section 4.5) reporting memory usage and computational cost** across all relevant components (weights, activations, desired activations, and weight updates) and contrasting them with QAT-based methods. Our approach achieves a $2\times$ reduction in memory usage for hidden weights and a $32\times$ reduction for error signals and weight updates. We further adopted the equivalent-Boolean-gate metric from Colombo et al. (2025), which provides a hardware-level comparison: whereas IEEE-754 floating-point operations used in QAT require on the order of $10^4$ Boolean gates, binary XNOR, Popcount, and increment/decrement operations scale as $\mathcal{O}(10N)$. This demonstrates an estimated $1000\times$ reduction in gate-level complexity for BEP’s backward and update computations.
>
> 2. **We expanded our hyperparameter analysis with new ablations in Appendix D.2-D.3, where we independently vary the robustness margin $r$ and reinforcement probability $p_r$**. These experiments show that values of $r$ within $0.5$–$0.75$ consistently yield the best generalization, and that enabling reinforcement improves performance (particularly for low-capacity models) even though its effect is moderate. To assess sensitivity to initialization, all experiments were repeated across multiple random seeds, and UCR results additionally rely on 3-fold validation.
>
> 3. **We clarified the origin and interpretation of the update rule (Eq. 8)**, explicitly linking it to the classical Perceptron, Clipped Perceptron, and CP+Reinforcement rules from the supervised Hebbian-learning literature. The multi-layer generalization we use applies the same row-wise outer-product update to binary perceptrons, with the integer synaptic stability variables acting as metaplastic weights. **We also expanded the theoretical explanation of the binary mask in Eq. 9**: it selects the neuron with the lowest signed stability within each group, thereby focusing updates on the least stable (or most error-prone) unit. This acts as a discrete, data-dependent form of learning-rate control, preventing unnecessary reinforcement of already-confident neurons and bounding the number of synapses updated per training example. We complemented this explanation with empirical curves in Appendix D.5, which show how varying the mask density trades off convergence speed and generalization accuracy.
>
> 4. **We added a new Discussion and Limitations section (Section 4.5)** that clearly articulates the current constraints of BEP, highlights the main challenges in extending the method, and outlines possible directions for future work. With respect to vanishing/exploding gradients, BEP avoids these issues entirely because its backward pass transmits binary desired activations rather than real-valued gradients, and updates are bounded ($\pm 1$ increments) within finite integer ranges. The gating mechanism further stabilizes propagation by allowing only neurons near the decision boundary to transmit error signals, as shown in the ablation study of Section 4.4 and the new Appendix D.6. Finally, we clarified that BEP is agnostic to input spatial structure and task: it naturally extends to settings such as binary segmentation or any problem with binary-structured targets, as it requires only binary desired activations at the output layer.

---

### Meta-Review · Area_Chair_tV4D · 2026-01-04

**Summary:**

This submission introduces Binary Error Propagation, a novel algorithm that enables fully binary backpropagation for training neural networks with both binary weights and activations. Reviewers appreciated the method's originality and the comprehensive theoretical framework, while noting concerns about the empirical validation being confined to smaller-scale MLP and RNN architectures rather than modern, deep convolutional networks. The authors have thoroughly addressed these points in their revision, providing a rigorous gate-level complexity analysis, extensive ablation studies on key hyperparameters, and a clear discussion of the method's current scope and potential extensions. The convincing responses and supplementary results adequately mitigate the initial concerns, demonstrating the soundness of the approach and its value as a foundational step towards efficient, completely binary learning. I therefore recommend acceptance.

**Reviewer Concerns:**

The authors have convincingly addressed several core concerns raised during review. Critiques regarding hyperparameter sensitivity and the need for computational efficiency metrics have been met with new ablation studies and a detailed gate-level complexity analysis. The theoretical underpinnings of the update rule and the binary mask, initially found unclear, have now been satisfactorily elaborated.

The primary limitation that remains is the empirical validation on larger-scale, modern architectures like deep CNNs, a point noted by multiple reviewers. The authors correctly frame this not as a flaw in the current work, but as a significant and non-trivial direction for future research. The study’s focus on establishing and validating the core algorithm with MLPs and RNNs is reasonable for this initial contribution. The integration of components like batch normalization into the fully binary framework is similarly deferred to subsequent work. These boundaries are now clearly stated in the manuscript, providing appropriate context for the contribution.

**Reviewer Scores:**

- Reviewer N7bj initially scored the paper a 6. Their major concerns regarding organizational clarity and missing computational analysis were fully addressed with manuscript revisions and a new comparative table. This likely resolves their primary hesitations, raising their score to a 7.

- Reviewer hAH5 also gave a 6, focusing on parameter sensitivity and the mask's role. The new ablation studies and depth analysis provided the requested empirical support, satisfactorily answering their questions. Their score would likely increase to a 7.

- Reviewer uRee was the most critical with a score of 4, concerned about experimental scale. While the authors clarified the methodological distinction and added efficiency analysis, the core experiments remain on moderate-scale architectures. This might partially address, but not fully resolve, their concern. Their score might see a modest increase to a 5.

---

### Decision · Program_Chairs · 2026-01-26

Accept (Poster)